# Resource Recovery and the Sherwood Plot

**DOI:** 10.3390/e25010004

**Published:** 2022-12-20

**Authors:** Georgios Karakatsanis, Christos Makropoulos

**Affiliations:** 1Department of Research, EVOTROPIA Ecological Finance Architectures Private Company (P.C.), 190 Syngrou Avenue, 17671 Kallithea, Greece; 2Department of Water Resources and Environmental Engineering, School of Civil Engineering, National Technical University of Athens (NTUA), 9 Heroon Polytechneiou St., 15870 Zografou, Greece

**Keywords:** resource recovery, Sherwood Plot (SP), Value Added Compound (VAC), entropy, dilution, recovery cost, cost structure, cost ontology, market formation process, market concentration, information entropy

## Abstract

Our work analyzes the biophysical and economic foundations of the *Sherwood Plot* (SP). In general, the SP depicts the theoretical relationship between the *cost of recovering* a target material or an identified *Value Added Compound* (VAC) from a waste matrix and its *dilution* in the waste matrix; specifically suggesting that the recovery cost is reverse proportional to the VAC’s dilution in it. We further utilize the SP as a scientifically consistent and economically coherent analytical framework for measuring *resource recovery* performance. Initially, we analyze the SP’s fundamental physical properties, as well as its many potential economic extensions. Specifically, we substantiate the relation between a VAC’s *Entropy, Dilution* and *Recovery Cost*. On these grounds we present the SP’s remarkable and numerous economic properties that make it consistent to its physical foundations; thus integrating concisely its physical and economic aspects and postulate a *generalized SP function*. We further test econometrically the validity of an SP based on both *deterministic* and *stochastic* real data from a small-scale industrial unit of polyphenols’ recovery from natural fruit juice production residual wastewater. In turn, based on the fusion of our theoretical argumentation and empirical findings we dive into the *epistemological* extensions of the SP. Specifically, we study how the recovery *cost structure* at the single industry level is revealed by the SP and can be useful for postulating cost structure *ontologies*. Cost ontologies are in turn useful as a *diagnostic* of the *formation process* of *VAC recovery markets* as well as their structure and concentration, defining the industrial shares when many industries operate in the recovery of the same VAC.

## 1. Introduction

The *Sherwood Plot* (SP) -after the Chemical Engineer Thomas K. Sherwood who originally formulated the method [1,2] is based on the core assumption that waste matrices are potential sources for the recovery of valuable materials, as an alternative to mining them from virgin ore deposits. Specifically, the original postulation of the SP depicted the relationship between the *required market price* of a target material to be recovered from a waste matrix and its *dilution* in the latter. In this context, the dilution was defined as *reverse concentration* at a logarithmic scale for smoothing extreme differences in scales of measurement (e.g., the comparison of very small fractions of the target material in relation to thousands of dollars of required market price). The formation of a set of coordinates by these two variables structured a map as decision support visualization for the recovery -or not- of the resource from the waste matrix [3].

Empirically, even though many target compounds or elements were found at very low concentrations (respectively very high dilutions) in waste matrices, it was still profitable to recover them due to their high natural scarcity, while others with very high concentrations were not, due to their very low market price that constituted their mining from virgin deposits to be a much more profitable solution. In any case, this rather profound macroscopic observation has deeper microeconomic extensions, affecting fundamentally the market structure and the allocation of market shares between industries operating in the recovery of the same materials, as we examine in Section 2 and Section 3.

However, the original form of the SP came with significant drawbacks. These drawbacks are the main motivation for our work that aims at reconsidering its foundations and revitalize its validity on more solid ground. The original SP form although it depicted the market price as a vital criterion for deciding to recover a material (or not), *it made indirect assumptions on a unified cost of recovery between industries*, which in reality differs significantly from industry to industry due to differences in technology [4], *R&D* programs, *Intellectual Property* (IP) rights (e.g., patents), learning curves and process engineering [5]. In addition, as the recovery of resources from wastes is at the core of what is called today *Circular Economy* (CE) -with *Industrial Symbiosis* (IS) being one of its major vehicles- the main quantitative issue is to implement *full cost-benefit accounting*. Full cost accounting means that besides the conventional industrial processing costs (e.g., capital, operational) two (2) types of *environmental costs*/*benefits* should be also included: (a) The cost/benefit of *avoided pollution* (e.g., in terms of CO2 emissions savings or conservation of ecosystem services value) and (b) The cost/benefit of *avoided virgin resources’ consumption* that leads to higher *in situ* availability [6,7]. In natural resource economics literature the above are considered as a *natural capital consignment* for future generations; also in line to the Brundtland Commission’s definition of sustainable development “*Sustainable is the development that meets the needs of the present without compromising the ability of future generations to meet their own needs*” with focus on *intergenerational natural resource justice* [8]. Such cost-accounting schemes can also be found in the standards of the *European Union’s* (EU) legislation on water resources [9]. Hence, under the EU’s economy’s re-structuring towards the CE, the cost and benefit of *natural capital recovery and conservation* is becomes a pillar of its ongoing industrial paradigm shift.

According to the above, our work introduces three (3) main diversifications that aim at enriching the SP and adapt it better to the modern needs of industrial symbiosis clusters. *Firstly*, we review and re-postulate the SP in terms of the *cost of recovering a target material* from a waste matrix and its *dilution* in the waste matrix. This approach allows us to explore the *microeconomic* extensions on the cost structure and answer questions such as “*what fraction of the total cost concerns constant costs and what variable costs?*” that further defines what we introduce as *cost ontology*. Upon this basis, we further build a structural framework on the formation of an *industrial symbiosis market*. Such a market consists of *at least two* industries (where *j* = number of industries; *j*∈N-[0, 1]) operating in the recovery of the target resource. In turn we study the market structure’s *properties* via selected *Market Structure Diagnostics* (MSDs) deriving from their market shares.

*Secondly*, we examine the relationship between *Entropy* and *Dilution*. Although the SP is a remarkably flexible model, even compatible to *Information Theory* approaches [3,10,11,12], for more conventional approaches concerning *Value-Added Compound* (VAC) dilutions in wastewater it is required to distinguish between the two concepts. Even if the SP can be utilized as an *entropy-cost* graph for specific cases that will be discussed (e.g., distribution of a VAC’s molecules residence in a compartmentalized volume), in our work the most definitive cost variable is the VAC’s *dilution* in the wastewater matrix. That is because we assume that the VAC is *always* at a *maximum entropy* (MaxEnt) state.

*Thirdly*, we apply this modified SP version to the recovery of a *reference VAC* as target material diluted in *wastewater*. In this context, we continue to adopt the original SP’s main theoretical assumption that the VAC’s dilution and its recovery cost from a wastewater stream are assumed to be *reverse proportional*; as the effort (cost) for recovering a desired target compound will increase if its concentration in the waste matrix is lower. Although the SP exists as analytical framework for more than 60 years, its application has been rather limited, which in part can be attributed to the fact that the CE as a *new economic paradigm* had not been a priority until recently.

## 2. Materials and Methods

### 2.1. Physical Foundations of the Sherwood Plot

The theoretical framework of the SP suggests that *as the statistical probability of finding the desired VAC within a wastewater matrix becomes lower, the cost of processing and recovering an economically viable mass becomes highe*r. Although the SP has remained relatively unutilized by waste recovery science, its potential utilization could offer significant insights and value for quantifying the performance of CE *markets*. One of our core tasks is to discuss the statistical features of the SP, which we identify to have intrinsic physical and economic meaning for building a concise analytical and modelling framework.

Primarily, it is necessary to elaborate on the original SP version, via the econometric relation connecting a VAC’s dilution in a wastewater matrix and its cost of recovery. Although in the initial SP formulation, the dependent variable was the VAC’s *required market price*, in the following sections we demonstrate that its substitution with the VAC’s *recovery cost* provides us with more conceptual and operational conveniences, without dismissing the importance of the observed market price data at a secondary level of (*Profit and Loss* (P&L)) analysis. In any case, we consider necessary to begin with the view of the SP’s initial conceptualization and formulation. Based on the above, the National Academy of Engineering [13] reproduced a basic logarithmic SP relationship with the use of cross-sectional data on a set of selected materials (both chemical compounds and elements) found in industrial and urban waste matrices.

The gathered data from empirical observations are depicted in a two-dimensional map, where an *Ordinary Least Squares* (OLS) linear regression is performed to identify the linear relationship with the minimum error -as the squared sum of deviations. As shown in Figure 1a, we would ideally have available data (a) *from many industries* and (b) *for a sequence of dilution levels* for the same VAC. Simply stated, for each VAC dilution level we should have a *distribution* or a *range* of required market prices, reflecting the different recovery costs, as each industry is quite different from the other in terms of cost efficiency, economies of scale, technology etc.. One significant drawback of the standard SP depicted in Figure 1b is that it lacks market representation, as it provides only one coordinate couple, considering it either as a weighted average or a benchmark value. In conclusion, a typical emerging question is *what a typical SP’s structure would be, had we studied various combinations of the dilution-price relationship for a single substance*. Even this approach indicates the reverse proportional relationship between dilution and required market price. Our work though, restructures the SP framework for higher market representation.

Although the SP could theoretically (and most probably in real-world cases) take any other nonlinear form (monotonic or not), the linear model -either deriving from a mathematical transformation (e.g., the log-log plot) or from a default data relation (as is in our case study)- is the most convenient to use, without excluding more complex depictions. The original form of the SP concerning a material’s required market price for every level of its dilution in a waste matrix is the following:(1)$$P\left(m_{i}^{-1}\right)=a_{i}+b_{i}\cdot \left(\frac{1}{m_{i}}\right)$$

In Equation (1), the required market price *P* of the *i*th material at dilution level *m*^−1^, depends linearly on the dilution level (*m_i_*^−1^), with two parameters; a constant factor *a* and a variable factor *b*. From Figure 1b it is obvious that while the majority of materials is concentrated within the range of *Dilution* (=0, 3) and *Market Price* (=0, 2) there are outliers that contain significant economic meaning. Specifically, we may distinguish the case of vanadium (*V*), where for even higher dilution levels (in relation to the main cluster) the required market price for its economic recovery is still very low. In contrast, the required market price of vitamin B-12 is very high in relation to its dilution level and in relation to the position of other elements. However, as already mentioned, it should be taken into account that the specific SP contains only a single combination of price-dilution coordinates for a set of highly diverse substances. Accordingly, we may consider that various SP depictions derive from very diverse waste matrices; meaning that the same waste can be recovered from different streams with cost differentiation. 

Respectively, as far as the SP technicalities are concerned, the scale differential between dilutions and required market prices may be significant (for instance the cases of Vanadium, vitamin B-12 and Radium in Figure 1b). Hence, due to the huge differences of scale between VAC dilutions and market prices, typically, SPs are depicted in *log*–*log* scale—although this is not the strict rule.

To define a substance’s dilution—as a core SP variable—we resort to the *International Union of Pure and Applied Chemistry’s* (IUPAC) definitions of substance concentrations. According to the IUPAC [14] standardization, the general formulation of a target VAC mass (*m_i_*) concentration (*ρi*) in a volume (*V*) is formulated as:(2)$$\rho_{i}=\frac{m_{i}}{V}$$

Equation (2) depicts the basic concentration relationship for a typical volume of 1 liter (*L*) of water. Even if many alternative metrics of concentration or density (such as molar concentration, mass fraction or molality that is more suitable for binary solutions [15,16,17] and more compatible to *information entropy* approaches to mixes) could be eligible for the SP, the formulation of a material’s typical concentration in water is considered more suitable for both the SP’s theoretical generalization and our empirical case. In addition, we further assume that all measurements take place in laboratory conditions that represent *ideal environmental conditions* for *Standard Ambient Temperature and Pressure* (SATP). Specifically, the SATP conditions assume a combination of an ideal temperature (*T*) of 298.15 K (25 °C; 77 °F) and of absolute pressure (*P*) of 1atm (101,325 Pa; 1.01325 Bar) [14]. Respectively, for mathematical convenience Equation (2) is transformed to:(3)$$\rho_{i}^{-1}=\frac{1}{\left(\frac{m_{i}}{V}\right)}=\frac{V}{m_{i}}$$

However, as the typical value of the water volume of 1 L is equal to 1 (=1), Equation (3) can be interpreted as the *residual concentration capacity* of the VAC within the wastewater matrix:(4)$$\rho_{i}^{-1}=\frac{1}{m_{i}}=m_{i}^{-1}$$

The economic interpretation of Equation (4) is that the unutilized concentration capacity is in fact an equivalent to the cost increase factor for a mix that for each dilution level is assumed to be found at *maximum entropy* (or maximum dispersion in the volume of the wastewater matrix). We consider the whole mix as an idealized pure substance, consisting solely of water and the VAC diluted in it, where the critical factor for the VAC’s recovery is its dilution in the water volume. Obviously, in reality, wastewater discharges contain numerous other substances as well; however, even for such cases, the SP formulation is still valid. For the sake of simplicity though, we assume that the mix consists of only two substances, water and the VAC.

#### 2.1.1. Entropy, Dilution and the Sherwood Plot

A frequent misconception concerns the attribution of a *causal relationship* between *entropy* and *dilution*, suggesting that *dilution derives from entropy increase*. However, these concepts have to be fundamentally distinguished, as they could be equivalent only within specific contexts, mainly coming from *Information Theory* [10,18]. The increasing dilution of any target substance just signifies a *lower average statistical probability* to find the substance within the mix. However, assuming that the solution is contained in a tank, the VAC’s mass could be concentrated in specific areas of the tank, as it occurs in the initial stages of its dilution when mixed or in supersaturated solutions when precipitated. Hence, we need to further assume *perfect miscibility* of the VAC within the water matrix, so that for any concentration between 0% and 100% (*ρ*∈(0,1)) the VAC is perfectly solved *homogenously* at every coordinate of the tank’s volume that contains the mix. This is equivalent to the statement that the VAC is found *volumetrically* at a *Maximum Entropy* (MaxEnt) state. Should we separate the tank’s volume in *n* parts (*n*∈N^+^), whatever part of the tank’s volume we seek to recover the VAC from, we will have the exact same average probability to find it. In short, *whatever the VAC’s dilution may be, miscibility ensures that the VAC is spatially (volumetrically) uniformly (MaxEnt) distributed*.

Besides the volumetric view, from a *mass distribution* perspective, with *Shannon Entropy* as a prevalent concept, we may schematically depict the distinction between dilution and entropy *maximizations*. It is obvious from Figure 2a that *Shannon Entropy H(X)* concerns the mass distribution between the two compounds forming the solution and gets a *maximum value* only for the case of *equal probability*, which in our example is 50%–50%. At this ratio, there is actually no prevalent substance in the solution. Any deviation from the case of equal probability, either in favor of the *solute* (VAC) or of the *solvent* (Water), signifies *lower entropy*, as one of the substances will become *prevalent* in the mix. Hence, *even the case of extremely high VAC dilution signifies that the entropy of the solution becomes lower* as water becomes extremely prevalent in the mix. In contrast, SPs described by a constantly monotonic function, either linear or non-linear (although more exotic discontinuous and non-constantly monotonic SP forms cannot theoretically be excluded), has nothing to do with the entropy of the mix, but only with the difficulty (in terms of effort and cost) to find and recover a unit of VAC. Hence, while the maximum cost for a VAC’s recovery is when the VAC’s dilution asymptotically approaches infinity (as a theoretical upper limit), its maximum entropy is observed only when the mix consists of the solvent and the solute in exactly equal parts.

According to the above, we adopt the formulation of *Renyi Entropy* [19] as a *generalization of Shannon Entropy*. From an *Information Theory* perspective, with Renyi Entropy we depict the composition of the mix via the probability to find a molecule from each compound forming the mix. Hence, the Renyi Entropy for a system consisting of discrete elements (here distinct molecules) is:(5)$$H_{q}=\frac{1}{\left(1-q\right)}\cdot \ln\sum_{i=1}^{n}p_{i}^{q}$$

For the asymptotic convergence of the value of parameter *q* = 1, the Renyi Entropy becomes the typical *Shannon Entropy* for discrete systems [19]:(6)$$\underset{q\to 1}{Lim}H_{q}=H(X)=-\sum_{i=1}^{n}p_{i}\cdot \ln p_{i}$$

According to Equations(5) and (6), by using *Information Entropy* we can interpret statistically not only mass concentrations in the mix but other concentration concepts as well, such as molar number or volume. This is feasible by adopting a specific stoichiometric encoding of the molecules, as the probability of them found in the mix, with other features that work as weighting factors (e.g., molecular weight) taken into account. Here, we model the substances forming the mix with semantics that distinguish them and apply the *Shannon Entropy* formula in a straightforward way for interpreting dilution for both simple and complex mixes. Hence, if there are *m_i_* elements (compounds) that form a mix of molecules in a finite tank (*M* = ∑m_i_; *m*∈N^+^), the *entropy maximization* of the mix occurs for the *exactly same probability* (equiprobability) to meet any of the mix’s elements at any specific co-ordinate of the tank’s volume. Alternatively stated, for a mix *M* consisting of *m_i_* different elements, each element appears only once in the mix. For this special case, where all elements have the same probability *p*(*m_i_*) to be found by a process of random selection, with *p*(*m*_1_) = *p*(*m*_2_) = … = *p*(*m_i_*) = *1/M*, Equation (6) becomes:(7)$$H{\left(M\right)}_{MAX}=-\ln\left(\frac{1}{M}\right)=-\ln p_{i}$$

Hence, it is not the entropy itself that increases the cost of a perfectly miscible VAC’s recovery from a wastewater matrix but its dilution in it. The MaxEnt state *sets the necessary conditions for the highest possible VAC recovery cost*, which is graphically translated to the positioning of any MaxEnt VAC SP higher than any other non-MaxEnt VAC SP for a common range of dilutions. From that point, it is the VAC’s dilution primarily impacting the cost. Respectively, we can imagine two identical VACs, either non-perfectly soluble or non-soluble at all, that are precipitated at the bottom of a water containing tank. While the entropy in both of the above VACs would not be maximized, if their dilution were to be extremely high in relation to a VAC found at MaxEnt but at very low dilution, their recovery cost would still be higher.

The above apply for any mix *M* consisting of a constant (non-changing) integer number of elements *m*. The only case where the *dilution* is proportional to *entropy* is when the number of elements *m_i_* forming the mix and having equal probability to be found in a *MaxEnt* mix increases. Alternatively stated, when conserving the *MaxEnt* property of the mix, the dilution of each element *m_i_* increases proportionally to the mix’s entropy across the increase of the sum of elements ∑m_i_ that form it (*M_i_ < M_j_*→∑*m_i_* < ∑*m_j_*→H(*M_i_*) < H(*M_j_*); *m*∈N^+^). According to Equation (7):(8)$$\rho_{i}^{-1}\propto H{\left(M=\sum m_{i}\right)}_{MAX}\propto \sum m_{i}$$

The economic interpretation of Equation (8) is that *the cost of a VAC’s recovery at MaxEnt state from a wastewater matrix depends on its scarcity*, defined as the statistical probability to find a VAC molecule out of the total number of VAC molecules that are homogenously distributed in the mix’s volume contained in a tank [20]. With a process of selection—irrespective of the specific coordinates of the tank in which the selection takes place—the higher the VAC’s dilution, the lower is the statistical probability to find one of its molecules. Combining Equations (4) and (8), we may postulate that for any mix *M* at *MaxEnt* state consisting of *m* elements (*M* = ∑*m_i_*), the dilution of any *m_i_* asymptotically converges to infinity with the increase of the number of elements:(9)$$\underset{\sum m_{i}\to +\infty }{\lim}\rho_{i}{}^{-1}=+\infty$$

Even if we begin from an *Information Theory* framework without taking into account the mix’s physical properties, Equation (9) is valid for a mix with constant mass, where the only variable is the number of the substances comprising it. With constant mass and monotonic increase of the number of substances up to infinity (as an upper theoretical limit), the only way to preserve its *MaxEnt* state by Equation (7) is to reduce the concentration (increase the dilution) of any substance *m_i_*. Respectively to Equation (9), we may denote that when the number of elements *m* approach the critical value 1, the mix consists in only one substance (there is practically no mix) with its dilution asymptotically converging to zero:(10)$$\underset{\sum m_{i}\to 1}{\lim}\rho_{i}{}^{-1}=0$$

As the concept of *scarcity* is the corner stone of the field of *natural resource economics*, this argument allows us to establish the conditions for the generalization of the SP’s utilization. Whatever process we may examine, such as VAC recovery from wastewater matrices, recovery of water of upgraded quality, mining target minerals from ore deposits or energy and CO_2_ recovery [12], the SP can be applied with general validity. Other more special considerations concern the *equivalent cost* between perfectly and non-perfectly soluble VACs with different entropy states and dilutions (i.e., the equivalent cost of the same VAC in two different tanks *A* and *B*, where in tank *A* the VAC is at *MaxEnt* state with lower dilution and in tank *B*it is in *non-MaxEnt* state but with a higher dilution). Via the analysis of such cases, we could design *equivalent entropy*–*dilution* charts, offering specific value and significance at the *process engineering* level, either for a single industry or an industrial ecosystem that primarily aims at maximizing final VAC recovery, where its participating industries perform synergies by allocating operations and resources. Such cases are briefly discussed—as they concern process engineering restructuring options—but their thorough analysis is beyond the scope of this work and is left for a future paper.

#### 2.1.2. Process Engineering and the Sherwood Plot

Whether we are examining an individual industry or an industrial collective, a vital aspect of the SP concerns the information on the *process engineering* sequence as part of full cost accounting [21,22,23]. In most cases, the physico-chemical processes of industries for managing wastes either for VAC recovery or environmental neutralization and safe discharge vary significantly, determining fundamentally the cost structure of an SP and hence an industry’s position in the VAC’s recovery market.

Specifically, we can identify two (2) pillars of an industry’s decision challenges on deciding the change of its process engineering sequence: The *first* pillar is to chart accurately its cost structure “hot-spots” with high weight on its total cost function within full cost-benefit accounting and pricing schemes [24]. In particular, industries could identify specific parts of their process engineering sequence suffering from high costs and that are susceptible to improvements via redesign, substitution or re-structuring. Such a case is discussed in the empirical part of our work, where the (daily) variable electricity cost has small but non-negligible contribution to the formation of the SP coordinates. With rising energy costs being the most relevant issue today [25] and with an additional global institutional pressure to decouple electricity generation from CO_2_ emissions, there are various resorts for amortizing the effect of extreme energy cost variability in the process engineering sequence, whether at the production or at the financial level [26,27]. To secure stable energy prices—especially in cases where electricity from renewable sources is an option [28]—today constitutes a priority for preventing the disruption of process engineering sequences and supply chains. 

The *second* pillar concerns an industry’s option to finance targeted technical upgrades, improvements or even full-scale innovations [29,30] via R&D. Such interventions aim at upgrading its process engineering sequence, eventually *shifting* the whole SP towards a completely new set of dilution–cost coordinates. As the level of technology varies significantly among industries, such decisions concern the acquisition of *competitive advantages* and higher market shares. Such an SP shift would be equivalent to either a total *technical improvement* [31], which would translate as lower VAC recovery cost for each dilution level or to a *mixed new chart* (e.g., technical upgrade for a set of dilution–cost coordinates and downgrade for another). However, even for the cases where industries have similar process engineering sequences, we may still identify significant deviations in VAC recovery cost functions. These are indications of cost efficiency differences, which is how well they manage to combine the elements of their equipment. On the other hand, for samples of industries with different process engineering sequences for the recovery of the same VAC, the differences in dilution–cost coordinates may even skyrocket, which can be an indication of the sector’s internal decentralization and competition level.

Hence, with selected process engineering interventions we may examine the cost differential in the SP dilution–cost relationship, either for an overview of the recovery of different materials [32] or the analysis of a single material [33]. From Figure 1b we may easily obtain that the cost differential is *negative* (the cost of recovery is zero even when the dilution is positive; or when the dilution is zero the recovery cost will be negative). Although this specific restructured sample [13] may lack specific data that would suggest otherwise, in the majority of cases (including our empirical case study) we observe a *positive* cost differential [34]. Its economic meaning is that even at zero dilution the cost of the VAC’s recovery is positive due to the fact that a share of the industry’s costs concerns *constant costs* that are paid whether the industry recovers any VAC amount or not. In contrast, the *variable* costs concern the costs paid for anything that has to do *directly* with the recovery process (e.g., electricity, chemicals etc.). How high each of these shares may be will define the scale at which each industry operates best in relation to other industries in the VAC’s recovery market and reveal its cost structure. According to the above, the SP provides the additional utility as a starting point for further endogenous optimizations at the single-industry level.

### 2.2. Institutional Shifts and the Sherwood Plot

The shift of consumption preferences towards specific materials signifies a fundamental technological re-direction towards the development of innovative process engineering sequences [30] for more efficient and economical recovery of critical resources. Such shifts usually come from a change in legislation that establishes the corporate accounting of environmental costs of virgin resources and -respectively- the benefits of recycled resources [7,9]. Indicative examples of current consumption preference shifts are the declared EU’s transition from fossil fuels to conventional renewables and nuclear energy, the use of bio-plastics and recyclable eco-friendly materials. These interventions direct the focus of industrial R&D towards inventing new technologies, change process engineering sequences and restructure supply chains for lowering the cost of resources’ recovery. Hence, when accounting for the environmental cost of virgin resources and the environmental benefit of recovered ones, cost differences tend to be mitigated or even eliminated. Such legally directed market shifts comprise a determinant of the recovery cost affordability across the SP among other factors. This discussion is not only important as a part of the SP fundamentals, but also relevant to current structural changes in the EU economy, due to its transition towards the CE [35,36,37] as a typical case of large-scale institutional shift.

Particularly, an indicative large-scale institutional shift is the *European Commission*’s (EC) decision to turn its economy into the first globally climate-neutral economy until 2050, by introducing the *European Green Deal* (EGD) [38] –practically the EU’s agenda on sustainable growth. In order to achieve this overarching goal, numerous actions are to be followed in various sectors, including -among others- construction, energy, transportations, food supply, biodiversity, ecosystem services valuation, where potential carbon tariffs should be appointed for member states not curtailing *Greenhouse Gas* (GHG) emissions at the rate set. In relation to the above, the EC has adopted the new *Circular Economy Action Plan* (CEAP) [39] as one of the major building blocks of the EGD. The EU’s transition to a CE will both reduce pressure on virgin resource extraction and ecosystem’s pollution, while shifting the focus of R&D programs on resource recovery towards related sectors [40]; further enhanced from recent concerns on imported energy and material supply security [25]. In short, as the CE expands beyond GHG emissions’ reduction, the introduction of resource recovery metrics and models becomes a pivotal issue.

According to the above, the EU’s CE markets are currently estimated to contain a value between 78.9–84.9∙10^9^ €; further analyzed in 6.9–12.9∙10^9^ € deriving from value-added generation from materials re-utilization and 72∙10^9^ € deriving from landfill cost diversion [35]. However, even this estimation could be considered rather conservative in the long term, as it can activate numerous sustainable growth factors via *spillovers*; such as energy cost reductions via efficiency increases, leading to fossil fuel substitution and GHGs. Additional macroeconomic *multipliers* would concern the creation of new scientific and business fields contributing to the improvement of the environmental footprint of EU-27 products and services. In particular, in the period 2016–2020, the *European Investment Bank* (EIB) released a financial plan of 2.7∙10^9^ EUR to co-finance CE projects in various sectors, while this amount will increase to 10∙10^9^ EUR by 2023 with the implementation of the *Joint Initiative for the Circular Economy* (JICE). A projected gain from the facilitation of CE direct and indirect investments is CO2 emissions neutrality by 2050, accounting for 3% global impact of due to energy and resource productivity efficiency increases. Respectively, it is projected that 2∙10^6^ climate-neutral related jobs by 2030 will have been created. In addition, a strategic goal is to establish the most suitable underlying indices [41,42,43] for building tailored finance instruments for eligible CE projects of higher environmental performance and sustainable value-added chains [44,45,46,47]. Hence, it is quite obvious that institutional shifts activate numerous large-scale economic multipliers [48] that in SP terms may shift completely VAC recovery costs towards lower levels, making the operation in such sectors a very profitable business.

### 2.3. Economic Foundations of the Sherwood Plot

The physical properties of the SP described in the previous sections set the fundamental economic conditions for charting VAC’s *recovery possibilities*, *optimizations* and *frontiers*. The core issue here is to depict *parsimoniously*, coherently and consistently the emergence of a single industry’s SP properties. By extending a single industry’s SP microeconomic properties to the macroeconomic level, we can assess the performance of recovered materials as *intermediate production factors* within *Input-Output* (I-O) as well as *Life Cycle Assessment* (LCA) frameworks to identify the optimal economic and environmental use of target materials among a range of possible utilizations [7,43,49].

#### 2.3.1. Microeconomic Foundations

As already demonstrated, the SP depicts the reverse proportional relationship between VAC dilution in waste water matrices and recovery cost, due to the increasingly lower statistical probability of finding the VAC within a waste matrix. The SP’s rationale can be replicated for the recovery of all types of materials and even generalized for all types of resources, including energy recovery and wastewater reclamation (as it is often a complementary product to VAC recovery). However, for the development of the microeconomic foundations of the SP we focus on material recovery as theoretical benchmark.

According to Figure 3 we may begin from any initial coordinate of VAC dilution and cost of recovery (*X*_1_,*Y*_1_). By default, this option creates four (4) distinct areas with specific microeconomic properties. The boundaries of these areas are determined by the lines *Y = Y*_1_ and *X = X*_1_ respectively ∀*X*,*Y*∈(0,+∞), with the combination (*X*_1_,*Y*_1_) as their unique intersection point. Each shift from the central coordinate (*X*_1_,*Y*_1_) to any of the four formed areas has a particular microeconomic meaning. The resultant of each possible dilution-cost coordinate shift can be described with the following options: (a) An upward shift across the line *X = X*_1_ (hence for *Y** > *Y*_1_*; X* = X*_1_) line signifies that the VAC’s recovery cost from the waste stream has increased for the same dilution level. A shift to the left across the *Y = Y*_1_ line (hence for *Y** = *Y*_1_; *X** < *X*_1_) signifies the same recovery cost but for lower dilution levels (which at the *X*_1_,*Y*_1_ coordinate depicted a higher dilution level). Any other coordinate falling into these boundaries and in the compartment in between (upper left) signifies relocation towards a *worse by a Pareto criterion performance* in the SP, with higher VAC recovery costs and lower dilutions in relation to *X*_1_,*Y*_1_; (b) In contrast, a downward shift across the line *X = X*_1_ line (*Y** < *Y*_1_; X* = *X*_1_) signifies that the VAC’s recovery cost from the waste stream has decreased for the same dilution level. Respectively, a shift to the right across the *Y = Y*_1_ line (*Y** = *Y*_1_; X* > *X*_1_) signifies the same VAC recovery cost but for higher dilution levels. Any coordinate falling into these boundaries and in the compartment in between (lower right) signifies relocation towards a *better by a Pareto criterion performance* in the SP, with lower VAC recovery costs and lower dilutions in relation to *X*_1_,*Y*_1_; (c) For the remaining area in the lower left compartment and (d) the upper right compartment, the resultant is indefinitive as all these coordinate sets consist of a trade-off between lower (higher) VAC dilutions and lower (higher) recovery costs. From a microeconomic standpoint, these areas crossed by a SP depict in principle the *reverse proportional relationship* (irrespective of its intensity) between VAC dilution and cost of recovery.

An additional proof that a typical SP (whether linear or non-linear) crosses through the trade-off areas (lower left, upper right) is that these compartments are the only ones providing a full chart of *infinite equivalent solutions* to the initial coordinate *X*_1_,*Y*_1_, so that the *per unit* (average) cost remains constant. In general, the average cost of VAC recovery at a benchmark dilution level *m_i_* is:(11)$$C_{i}\cdot m_{i}^{-1}=\bar{C}$$

By Equation (11) to maintain a constant average cost, the total recovery cost *C_i_* must be increasing across increasing dilution, as depicted in Figure 3b. Respectively, the ratio between the VAC’s average cost of recovery and its total cost of recovery must always depict the dilution level:(12)$$m_{i}^{-1}=\frac{\bar{C}}{C_{i}}$$

Equations (11) and (12) depict the chart of equivalent coordinates between dilution and recovery cost from two different sides; from Equation (11) the chart formed in Figure 3b resembles to typical *Indifference* (for consumption) or *Isoquant* (for production) *Curves* with strict convexity [50]. From Equation (12) the gap between the average and total cost of recovery depicts the *residual capacity*, interpreted as the unutilized capacity of the solvent for additional VAC concentration until its saturation that works as a *recovery cost multiplier*.

Specifically, for 100% VAC concentration, the average and total recovery costs coincide, as it has already been discussed briefly in Equations (3) and (4). Finally, regarding the shifts of either some coordinates or the whole SP, besides the areas able to provide a straightforward conclusion on Pareto improvements (lower right) or deteriorations (upper left), the combinations within areas of an indefinitive resultant (lower left, upper right) would require more specific trade-off metrics (e.g., recovery cost elasticity to dilution) to provide us with more concise information.

#### 2.3.2. Macroeconomic Foundations

The micro-statistical (entropy) and microeconomic properties of the SP further emerge at the macroeconomic level via the direct connection of the concept of *entropy* to *resource scarcity* [51]. Roegen’s postulation begins from the *2nd Law of Thermodynamics* that dominates energy transformation processes via their *irreversibility*. With the classical definition of entropy by Clausius, a part of the thermal energy released from chemical bonds will turn into entropy, further unable to produce thermo-mechanical work. Roegen’s seminal ideas introduced a *paradigm shift* in the field of natural resource economics that until then had not realized the direct connection of this fundamental physical law to *economic scarcity* as the conceptual corner stone of economics.

Based on this reality, many *national accounting* models have either been restructured or designed from zero-basis to reflect the cost of natural resources and ecological carrying capacities [7] depletion per unit of economic product. These models further incorporate a variety of suggested indices [52], as integral part of the scientific discussion on the accurate full cost-benefit accounting. In addition, whatever the microeconomic peculiarities of single industries may be, at the collective (market or social) level a major macroeconomic challenge concerns organized resource transitions. Such cases are the *Kuznet’s Curve* [53] related to a society’s *learning* ability to shift its industrial processes towards higher ecological performance [54]. While the SP’s microeconomic utilizations concern process engineering sequences, its macroeconomic extensions concern the building of self-sustained CE markets via *NEXUS Energy-Water-Food* approaches, where recovered materials are *intermediate production factors* within *Input-Output* (I-O) frameworks [49].

#### 2.3.3. The Recovery Cost vs. the Required Market Price Approach

As already mentioned, one of our work’s pillars concerns the SP’s repostulation to depict the dilution-cost relationship in contrast to the original SP form on the dilution-required market price [2,55]. Although both contexts are generally valid -each containing specific advantages and disadvantages- we argue in favor of the dilution-cost relation as more straightforward, parsimonious, fundamental and with less incorporated uncertainties concerning complex market dynamics; hence suitable for both individual industries and industrial symbiosis clusters.

According to Figure 4b a SP is also a sub-chart gathering the subset of optimal dilution-cost coordinates from the set of all possible coordinates formed by all equivalent per unit cost charts to which the SP comprises the common tangent. For instance, in Figure 4b except for the common coordinate (*X*_1_,*Y*_1_) that depicts a *pivotal equivalent solution*, all other coordinates of the *X_i_∙Y_i_* chart are positioned higher than those of the SP for the same segment, depicting higher recovery costs for the same dilution levels. Hence, for any set of dilution values *X_i_*, where ∀*X_i_*∈(0,+∞), the SP comprises the *Pareto optimal* chart of the minimum recovery costs.

Materials present at high dilutions are recovered at a high cost. Another alternative interpretation of this reality from a commercial standpoint is that a material’s recovery has a *higher probability* to be considered a more expensive option in relation to its virgin ore extraction. We are denoting “higher probability” here, as the final decision on the VAC’s bulk supply source partially depends on its recovery cost by the single industry, partially on the virgin ore extraction cost -provided that full cost accounting is applied [7]- and partially on the VAC’s market price which is usually fluctuating. In the case where everything is accounted for and the recovery cost is still found to be the highest possible option, the material will be preferred to be mined by a virgin ore.

However, the recovery cost comprises the *fundamental unit* of determinants constituting the market price but not the only critical variable. The main argument for the priority of the SP’s cost-dilution approach is that the market price consists of many fundamental building blocks that already contain cost aspects, hence requiring further decomposition. In addition, the cost-dilution approach allows industries to quickly identify process engineering opportunities in advance and steer technological shifts towards lower recovery costs before the prices of virgin materials become prohibitively high due to irreversible resources exhaustion.

In addition, although the mechanism of prices may work efficiently in the short-term, the long-term stress on natural capital could be so high that even market prices could become unable to depict it adequately. In short, there exist additional reasons to choose the cost over the price approach. The most fundamental and crucial one is that market prices have a tendency to suffer from *endogeneity*. That is, they are affected *indirectly* by many factors determining it at a more fundamental level. Besides that, there are numerous other factors, such the VAC’s changing demand, the existence of potential substitutes and even speculators’ activity, especially when the VAC is traded in an exchange or if the market concentration for the specific sector is very high, making competition very low. In conclusion, the high price of a recovered material does not necessarily mean that it has a high cost of production, as this may be due to systematic speculation activity. In the context of industrial symbiosis, the primary discussion concerns the cost of recovering a resource material and secondarily all other factors that shape the material’s final market price.

### 2.4. Mathematical Formulation of the Sherwood Plot

In this section we develop the mathematical framework of the SP, both in its generalized theoretical and empirical forms. The empirical case reduces to the standard linear model based on available data from a very small-scale unit for polyphenols’ recovery, in the role of the VAC, from industrial wastewater. For the theoretical framework, we utilize the *Multinomial* and *Binomial* theorems to depict the analytical cost structure of an industry’s SP via a bottom-up approach, allowing all levels of granularity, from how effectively cost elements manage to combine to the total cost.

The cost structure derives from four (4) potential classifications as combinations of two main cost categories with two cost sub-categories each. The main categories are (1) *Constant Costs* and (2) *Variable Costs*, while secondary classes are (a) *Capital Expenses* (CAPEX) and (b) *Operational Expenses* (OPEX). A frequent misconception found in the literature is that “constant costs” are identical to “CAPEX” and respectively that “variable costs” are identical to “OPEX”. From an ontological perspective, there has to be a hierarchical distinction between these terms. Empirically, we may indeed verify a strong relation between these pairs; however, the modern finance world has developed a huge variety of sophisticated instruments, agreements and contracts that allow significant flexibility of cost management profiles. The line is of course quite thin and the potential to make combinations practically inexhaustible. 

In this context, we may define the *Constant Cost* as a cost that an economic unit (in our case the industry) *bears irrespective of the level of its production*. In simple words it has to pay it irrespective of whether it produces zero or its maximum volume. Respectively, *Variable Costs* are costs that are directly *related and proportional to the production volume of the economic unit*. This approach allows us to consider abstractly these categories as the main ones. At a secondary level, we have the *CAPEX*, defined *as any expense that concerns the creation of the infrastructure that will host the production process* (in our case the VAC’s recovery) and the *OPEX* as *any expense that is related to the production process itself*. These classifications are directly related to the *cost ontologies* that will be more thoroughly described in the next sections and are indicatively described in Table 1 below:

From their combinations and examples presented in Table 1, we understand that any CAPEX and OPEX type can fall into any category of constant or variable costs. For instance, a CAPEX investment for purchasing land or building an industrial unit that is financed by a bank loan via an agreement to repay it in 10 years with a fixed discount rate is a *constant CAPEX* as it is not directly related to production volume. It is also implemented in recurrent and predictable payments that provide the economic unit with high predictability and allow it to program in advance its expenses; thus, its annual balance sheets that have a fundamental impact to its shareholders and stock value. Contrarily, to repair a damaged equipment via hiring external personnel at short-notice in market prices is a *variable OPEX* as on the one hand it is directly related to the production process and on the other it was an emergency that could not be predicted in advance.

The potential to make combinations for building the optimal cost structure is practically inexhaustible, but outside the strict scope of our work; although, we dedicate a part of Section 3.2 for a brief discussion of the matter. In any case, the bottom line is that the cost structure of an industry engaged to a VAC’s recovery has definitive importance for its *potential market dominance* when issues of competition are introduced, as we thoroughly later analyze in Section 3.3.

The above approach provides us with a theoretical basis to identify an industry’s SP cost ontology (and as we argue in Section 3.3 of the market as well). In short, we utilize the properties of the *Multinomial Theorem* as a parsimonious approach to the industry’s *production cost function*, assuming that we have no other prior information on how efficiently the various elements of the industry’s infrastructure and, hence, cost factors collaborate together; or how each of them responds to exogenous changes (e.g., the change of electricity wholesale prices). With the combination of assumptions adopted above and the quantitative framework developed below, we are able to identify in deep detail an SP’s *cost ontology*. The generalized SP function is:(13)$$C\left(m_{ij}^{-1}\right)={\bar{A_{ij}}}^{1/\beta_{ij}}+B_{ij}{}^{1/\gamma_{ij}}\cdot {\left(\frac{1}{m_{ij}}\right)}^{1/\delta_{ij}}$$

In Equation (13), *C*(*m_ij_*^−1^) is the total cost of recovering a specific *i* VAC by a specific *j* industry, *A_ij_* (capped) is the sum of all *constant* (hence the cap to denote them) costs (both CAPEX and OPEX), *B_ij_* is the sum of all *variable* costs (both CAPEX and OPEX), 1*/m_ij_* is the VAC’s dilution as explained in Equation (4), while *β*, *γ*, and *δ* (with *β*,*γ*,*δ*∈R^+^) are *cost scaling factors*, denoting the type of *economies of scale*; *negative* (for *β*,*γ*,*δ*∈(0,1)), *constant* (for *β*,*γ*,*δ* = 1) and *positive* (for *β*,*γ*,*δ* > 1). In general, constant costs are by default inelastic and variable costs are easier to reduce. Hence, the above analytical framework allows the identification of cost factors that contribute most to the VAC’s total recovery cost, so that targeted cost reduction measures at the *process engineering level* can be adopted in order to shift the industry’s SP towards a more competitive market position.

To further analyze Equation (13), we may write variable *A_ij_* (capped) as the sum of all *n* constant costs:(14)$$\bar{A_{ij}}=\sum_{i=1}^{n}a_{i}$$

Respectively, we may follow the same rationale to write variable *B_ij_* as the sum of all *p* variable costs:(15)$$B_{ij}=\sum_{i=1}^{p}b_{i}$$

However, according to Table 1, constant and variable costs can be further classified into CAPEX and OPEX; if we define *n* as the sum of all constant costs, we may further classify this sum into *n_C_* and *n_O_*, with *n_C_* depicting the sum of *constant CAPEX* and *n_O_* the sum of *constant OPEX* costs:(16)$$n=n_{C}+n_{O}$$

Respectively, we may follow the rationale of Equation (16) for the sum *p* of variable costs:(17)$$p=p_{C}+p_{O}$$

According to Equation (16), the full deployment of *A_ij_* (capped) as the sum of constant costs is written as:(18)$$\bar{A_{ij}}=\bar{A_{ij}^{C}}+\bar{A_{ij}^{O}}=\sum_{i=1}^{n_{C}}a_{i}^{C}+\sum_{i=1}^{n_{O}}a_{i}^{O}$$

Respectively, we may write *B_ij_* to be the sum of all variable costs:(19)$$B_{ij}=B_{ij}^{C}+B_{ij}^{O}=\sum_{i=1}^{p_{C}}b_{i}^{C}+\sum_{i=1}^{p_{O}}b_{i}^{O}$$

Equations (18) and (19) are in full accordance to the cost ontologies’ classification presented in Table 1 and can be utilized for a very analytical depiction of the following mathematical framework concerning the interrelations of all cost elements within a *Computable General Equilibrium Model* (CGEM) [56]. CGEMs are mainly based on a matrix approach of the various inputs and outputs of production functions that can be dynamic, with flexibility to both incorporate new cost elements at each time step (costs elements that did not exist in previous time steps) and to reallocate existing cost elements across the 4 classes in Table 1 without the need of *a priori* static assumptions on the mathematical form of an economic unit’s cost function that usually have limited temporal validity.

#### 2.4.1. Generalized Theoretical Formulation

Having defined the cost structure space in Table 1 and Equations (13)–(19), we utilize the properties of the *Multinomial* and *Binomial* theorems to generalize the SP’s mathematical formulation. Specifically, we model each of the above four cost classes with the Multinomial Theorem and then encapsulate them within the two main categories (constant, variable) in a Binomial Theorem framework. This structure allows us: (a) to conceptualize cost elements parsimoniously *a priori* in the simplest possible form as a sum with (b) preserving the ability to detect more complex relations (e.g., synergies) between them via econometric (empirical) analyses and (c) to be consistent with *neoclassical* economic production functions (e.g., generalized power, CES, Cobb-Douglas) as their generalized form.

Additionally, this approach allows the logarithmic transformation of econometric models in order to study at every time step *t* the statistical significance of *autocorrelations* and *cross* correlations between CAPEX and OPEX, as well as their multiplication effects per level of VAC dilution (positive, negative or constant economies of scale), treating each industry’s process engineering as a CGEM.

According to Equations (14), (16) and (18), we may model the constant costs’ function—either CAPEX (*C*) or OPEX (*O*)—depicted by the variable *A_ij_* (capped) according to the *Multinomial Theorem* as:(20)$${\bar{A_{ij}}}^{1/\beta_{ij}^{C,O}}={\left(a_{1j}+a_{2j}+\ldots +a_{nj}\right)}^{1/\beta_{ij}^{C,O}}$$

Equation(20) depicts the sum of all *constant* cost elements, separately per class (CAPEX, OPEX) with an exponent 1*/β_ij_* (with *β*∈R^+^) that respectively to parameter *δ_ij_* is a *cost scaling factor*, denoting the type of *economies of scale*; *negative* (for *β*∈(0, 1)), *constant* (for *β* = 1) and *positive* (for *β* > 1). In this case, the main difference from *δ_ij_* is that *β_ij_* comprises a measure of how well the constant cost elements work together in order to achieve a *cost saving scaling* effect. Typical examples is the discount rate reduction if the industry signs an exclusive agreement with a specific banking institution for financing the purchase of all of its tangible assets or sign an exclusive collaboration with a legal firm for managing its intangible capital (e.g., intellectual property (IP)). Respectively, we may follow a similar rationale for the variable costs (CAPEX, OPEX), depicted by variable *B_ij_*:(21)$$B_{ij}{}^{1/\gamma_{ij}^{C,O}}={\left(b_{1j}+b_{2j}+\ldots +b_{pj}\right)}^{1/\gamma_{ij}^{C,O}}$$

In Equation (21), the exponent *1/γ_ij_* has the same functionality as exponent *1/β_ij_* on the economies of scale achieved from how well the *variable* cost elements of each class collaborate together. Respectively, we may consider as a typical example the combined cost saving effect from the combination of a product certification and marketing project or the cost saving effect from the *ad-hoc* training of temporary manual labor personnel.

In overall, the combination of the Multinomial and Binomial theorems into economic models is better from typical multiplicative production functions (e.g., the Cobb–Douglas) as it allows for specific cost elements to take zero value (=0) without collapsing the model. In contrast, in neoclassical production functions -even in some CGEM types- this is not possible as it will cancel out the whole function even if a single element takes zero value. In a few words, this model structure allows for both multiplicative and additive effects even when some cost elements take zero value. In more technical terms, the *Multi-Bi* modeling approach achieves to maintain the properties of *substitutability* and *complementarity* of cost elements. *Substitutability* concerns the degree at which the cost elements can be mutually substituted (e.g., converting a fixed rate loan to variable rate), where when a production factor becomes extremely scarce its substitutability converges asymptotically to zero, while its complementarity to infinity. *Complementarity* of cost elements and production factors concerns their constraint to necessarily work together at minimum ratios. A typical example of such a case would be the minimum amounts of chemicals required by the related equipment, according to its technical specifications to achieve the desired VAC’s recovery found at specific dilution in the wastewater matrix.

According to the above, the formulation of Equations (20) and (21) relies on the exponentiation property that allows us to express any positive real number *y* (with *n*∈R^+^) as an exponentiation combination of any two (or more) positive real numbers *x* and *n* (with *x*,*n*∈R^+^) that can generate number *y* with various combinations of natural, rational and irrational numbers for the base and the exponent (e.g., the base rational and the exponent irrational, the base irrational and the exponent natural, both irrational, etc.):(22)$$x^{n}=y$$

From an econometric perspective, the logarithmic linear transformation of Equation (22) is:(23)$$\log_{x}y=n$$

Additionally, as the exponents are not necessarily natural numbers (positive integers), we resort to the *Gamma Function* (*Γ(x)*) that extends the properties of factorials to all real (natural as well as negative and positive non-integers) and complex numbers. However, in our case we only use the domain of positive real numbers (natural and non-integer) as a more suitable (for industrial processes) subset of the above spaces. Hence, we may write:(24)z!=Γ(z+1)

According to the properties of the Multinomial Theorem, the general analytical form for unifying Equations (20)–(22) is:(25)$${\left(\sum_{i=1}^{n}x_{i}\right)}^{1/\lambda }=\sum_{\sum_{i=1}^{n}1/\lambda_{i}=1/\lambda ;\lambda_{1},\lambda_{2}\ldots ,\lambda_{n}\in [0,+\infty )}\left(\begin{array}{c} 1/\lambda \\ 1/\lambda_{1},1/\lambda_{2},\ldots ,1/\lambda_{n} \end{array}\right)\cdot \prod_{i=1}^{n}x_{i}^{1/\lambda_{i}}$$

As part of Equation (25) for any positive integer value of the exponents 1*/β_ij_* and 1*/γ_ij_* (*1/β*; 1/*γ*∈Ν^+^) in Equations (20) and (21), the *Multinomial Coefficient* as part of Equation (25) is written as:(26)$$\left(\begin{array}{c} 1/\lambda \\ 1/\lambda_{1},1/\lambda_{2},\ldots ,1/\lambda_{n} \end{array}\right)=\frac{\left(1/\lambda \right)!}{\left(1/\lambda_{1}\right)!\cdot \left(1/\lambda_{2}\right)!\cdot \ldots \cdot \left(1/\lambda_{n}\right)!}$$

However, for the special case where the exponents 1/*β_ij_* and 1/*γ_ij_* give positive non-integer values (1/*β*; 1/*γ*∈{[R^+^] − [Ν^+^]}), according to Equation (24) the Multinomial Coefficient is written [57] as:(27)$$\left(\begin{array}{c} 1/\lambda \\ 1/\lambda_{1},1/\lambda_{2},\ldots ,1/\lambda_{n} \end{array}\right)=\frac{\Gamma \left(1+\sum_{i=1}^{n}1/\lambda_{i}\right)}{\prod_{i=1}^{n}\Gamma \left(1+1/\lambda_{i}\right)}$$

As the Gamma Function can be used for both integer and non-integer exponent values, we may write again Equation (25) in its more general form as:(28)$${\left(\sum_{i=1}^{n}x_{i}\right)}^{1/\lambda }=\frac{\Gamma \left(1+\sum_{i=1}^{n}1/\lambda_{i}\right)}{\prod_{i=1}^{n}\Gamma \left[1+1/\lambda_{i}\right]}\cdot \prod_{i=1}^{n}x_{i}^{1/\lambda_{i}}$$

According to Equations (24)–(28), we may re-write Equation (20) by separating the cost elements of the constant costs’ category to its two (2) distinct classes (CAPEX, OPEX) into an integrated *Multi-Bi* model for exploring the complete topology of causal cross-relationships between constant CAPEX and constant OPEX:(29)$${\bar{A_{ij}}}^{1/\mu_{\bar{A}}}={\left({\bar{A_{ij}^{C}}}^{\left(1/\beta_{ij}^{C}\right)}+{\bar{A_{ij}^{O}}}^{\left(1/\beta_{ij}^{O}\right)}\right)}^{1/\mu_{\bar{A}}}$$

In Equation (29), each constant cost class (CAPEX, OPEX) is modeled both separately by the Multinomial Theorem and cross-dependently, according to the properties of the *Binomial Theorem*. In this case, the economic logic behind this upscaling approach is to identify *how well the constant CAPEX elements combine to the constant OPEX elements to achieve economies of scale*; expressed by the exponent 1/*μ_A_* (with *μ*∈R^+^) that, respectively, to parameters *β_ij_*, *γ_ij_* and *δ_ij_* is a *cost scaling factor*, denoting the type of *economies of scale*; *negative* (for *μ*∈(0, 1)), *constant* (for *μ* = 1) and *positive* (for *μ* > 1). Typical examples of such cross-section dependencies would concern how the working space (e.g., a testing laboratory) is optimized in terms of ergonomics or location proximity to other facilities in order for the R&D personnel to maximize its productivity and performance or how the option of paying internal maintenance personnel combines to the existing insurance contract on the equipment. Respectively, we follow the same rationale for CAPEX and OPEX of variable costs:(30)$$B_{ij}{}^{1/\mu_{B}}={\left(B_{ij}^{C}{}^{\left(1/\gamma_{ij}^{C}\right)}+B_{ij}^{O}{}^{\left(1/\gamma_{ij}^{O}\right)}\right)}^{1/\mu_{B}}$$

In Equation (30), the exponent 1/*μ_B_* has the same domain range and rationale (economies of scale) to 1/*μ_A_*. A typical example of such a case could be the synergy between marketing expenses and travel costs for personnel presence in conferences and exhibitions. Based on the *Binomial Theorem* [58], the general analytical form for integrating Equations (29) and (30) is:(31)$${\left(x_{1}+x_{2}\right)}^{1/\mu }=\sum_{k=0}^{n}\left(\begin{array}{c} 1/\mu \\ k \end{array}\right)\cdot x_{1}^{\left(1/\mu \right)-k}\cdot x_{2}^{k}$$

Respectively, to Equation (26) on the Multinomial Theorem, for any positive integer value of the exponents 1/*μ_A_* and 1/*μ_B_*, the *Binomial Coefficient* as part of Equation (31) can be written as:(32)$$\left(\begin{array}{c} 1/\mu \\ k \end{array}\right)\begin{array}{ccc} = & \frac{\left(1/\mu \right)!}{k!\cdot \left[\left(1/\mu \right)-k\right]!}, & k\in [0,n)\\ = & 0, & k\in R-[0,n) \end{array}$$

However, for the special case where the estimation of exponents 1/*μ_A_* and 1*μ_Β_* result to positive non-integer values (1/*μ*∈{[R^+^] − [Ν^+^]}), we may again use the properties of the Gamma Function as:(33)$$\left(\begin{array}{c} 1/\mu \\ k \end{array}\right)=\frac{\Gamma \left[\left(1/\mu \right)+1\right]}{\Gamma \left(k+1\right)\cdot \Gamma \left[\left(1/\mu \right)+1-k\right]}$$

As the Gamma Function can be used for both integer and non-integer exponent values, we may write again Equation (31) in its more general form as:(34)$${\left(x_{1}+x_{2}\right)}^{1/\mu }=\frac{\Gamma \left[\left(1/\mu \right)+1\right]}{\Gamma \left(k+1\right)\cdot \Gamma \left[\left(1/\mu \right)+1-k\right]}\cdot x_{1}^{\left(1/\mu \right)-k}\cdot x_{2}^{k}$$

According to Equations (14)–(34), the generalized SP form of Equation (13) contains up to seven (7) parameters (depending on desired granularity) and has the following analytical form:(35)$$C\left(m_{ij}^{-1}\right)={\left({\bar{A_{ij}^{C}}}^{\left(1/\beta_{ij}^{C}\right)}+{\bar{A_{ij}^{O}}}^{\left(1/\beta_{ij}^{O}\right)}\right)}^{1/\mu_{\bar{A}}}+{\left(B_{ij}^{C}{}^{\left(1/\gamma_{ij}^{C}\right)}+B_{ij}^{O}{}^{\left(1/\gamma_{ij}^{O}\right)}\right)}^{1/\mu_{B}}\cdot {\left(\frac{1}{m_{ij}}\right)}^{1/\delta_{ij}}$$

A special case of Equation (35) would be the one of *Constant Elasticity of Substitution* (CES) [59], where the per cent change in the ratio between the quantities of two cost elements that can substitute each other to the per cent change in their costs remains constant for every combination. In short, CES denotes that per cent substitutability between cost elements remains constant. For constant costs (CAPEX, OPEX), the conditions for CES according to Equation (35) would be for:(36)$$\beta_{ij}^{C}=\beta_{ij}^{O}=\beta_{ij}=\frac{1}{(\frac{1}{\mu_{\bar{A}}})}$$

For instance, *ceteris paribus*, a per cent increase of the variable discount rate on a loan would provide an incentive to the industry for converting a fraction of its loans from variable to fixed discount rate. For CES, *the per cent change of the industry’s fixed and variable discount rate loans ratio would reflect exactly the per cent change between fixed and variable discount rates*. For the *variable* costs (CAPEX, OPEX), the conditions for CES by Equation (35) would, respectively, be:(37)$$\gamma_{ij}^{C}=\gamma_{ij}^{O}=\gamma_{ij}=\frac{1}{\left(\frac{1}{\mu_{B}}\right)}$$

In addition, we set the conditions of each parameter for the cost function according to the above approach to be economically coherent and mathematically consistent:(38)$$\beta ,\beta_{i},\gamma ,\gamma_{i},\delta_{i},\lambda ,\lambda_{i},\mu ,\mu_{i}\in \left(0,+\infty \right)$$

As far as the main variable is concerned, the VAC’s concentration fraction ranges asymptotically from 0–100%, as described by Equation (2):(39)$$m_{ij}\in \left(0,1\right)$$

From Equation (39), we may write Equation (40), which in economic terms means that when the VAC’s concentration asymptotically approaches zero (=0), the cost of its recovery is asymptotically approaching infinity. This result was first implied in Equation (3). However, we may fairly consider this as a theoretical upper limit, as it most probably a technical limit on the industry’s capacity to recover the VAC will have been imposed for a much higher concentration than zero:(40)$$\underset{m_{ij}\to 0}{Lim}C_{ij}=\infty$$

In contrast to Equation (40), as the VAC’s concentration approaches 100%, its recovery cost is minimized and consisting of the sum of all constant (capped) and variable cost categories. As discussed in Section 2.1.2, this is the *positive cost differential* with the condition of minimum VAC recovery cost/effort:(41)$$\underset{m_{ij}\to 1}{Lim}C_{ij}=\bar{A_{ij}}+B_{ij}$$

In addition, the exponents of economies of scale *β*, *γ*, and *μ* for each cost class and category can affect the VAC’s total cost of recovery, as depicted in Equations (20) and (21) and Equations (29) and (30). Specifically, when the value of the exponents approaches zero, the VAC’s cost of recovery approaches infinity *irrespective of the VAC’s concentration*. The economic rationale behind Equation (42) is that the cost elements in all categories combine in an extremely inefficient way so that the VAC’s cost of recovery skyrockets. Specifically:(42)$$\underset{\beta ,\gamma ,\mu \to 0}{Lim}C_{ij}=\infty$$

In contrast to Equation (42), when the values of the exponents approach infinity, the VAC’s recovery cost becomes zero (=0) *irrespective of the VAC’s concentration*. In that case cost elements combine in an extremely efficient way so that even extremely low concentrations of the VAC become irrelevant for its recovery. Respectively, to the rationale of Equation (40), this is a rather theoretical case. Additionally, this is the only case where the cost of recovery may be equal to zero as the *universally minimum cost of recovery*:(43)$$\underset{\beta ,\gamma ,\mu \to \infty }{Lim}C_{ij}=0$$

The conditions described in Equations (42) and (43) are boundary and contain a rather theoretical value than empirical one. In overall, the combination of the Multinomial and Binomial theorems for a unified cost CGEM-type cost function serves multiple purposes for *industries that prioritize the detailed charting of their microeconomic and process engineering causalities*. An initial summation formula transforms into a product; hence, an industry may explore all possible combinations between its cost function parameters.

From a quantitative perspective the above are related to the econometric investigation on which multiplicative relations “survive” econometric testing as statistically significant from a space of possible combinations, without “a priori” assumptions that may be based on philosophical or other biases. Or—alternatively—in the best case just be a special case of a more generalized model, exactly as the Cobb–Douglas production function is a special case of transcendental forms [60]—that in turn are not entirely abstract, inclusive or based on the principle of “parsimony” but on initial assumptions and desired properties.

In addition, besides examining the coefficients for each element of the same cost class (CAPEX, OPEX) that belong to the same cost category (Constant, Variable) via the Multinomial Theorem to estimate their efficiency (or deficiency) to form “collaborative” endogenous relationships, the use of the Binomial Theorem follows the same rationale; only this time concerning at a the cross-class (CAPEX→OPEX, OPEX→CAPEX) collaborative relationships to measure the *scaling* of synergistic effects between cost elements of different categories. Hence, irrespective of the frequent simplicity of empirical SPs (usually standard linear with two parameters) the above general formulation is inclusive of all possibilities of the VAC’s recovery total cost function.

#### 2.4.2. Prevalent Empirical Formulation

The generalized SP modeling can yield an extremely analytical and complex set of relationships (depending on the desired granularity) between the parameters (expressed by the cost coefficients). However, the most frequent empirical form, including the original one formulated by T.K. Sherwood [2] as well as by later authors and irrespective of whether the adopted methodological framework concerned physical properties [3,13,33] or Information Theory approaches [10,11], is the linear one. Whatever specific form an SP may take, as long as it is expressed by a monotonic function (linear or nonlinear), the principles for distinguishing the main cost categories (Constant, Variable), as well as the derivative ones (total, marginal) remain the same, as presented in Figure 5 below:

For the special case of exponents’ values in Equation (38) *β = γ = δ = μ =* 1, where no cost scaling effects exist, the generalized SP model of Equation (35) reduces to the *Standard Linear Model* (SLM) that is described in Equation (1), with the difference that the depended variable is the industry’s total cost of VAC recovery instead of the VAC’s required market price.

#### 2.4.3. Cost Structure Analytics

For the SLM we may use the VAC dilution–cost coordinates to perform an *Ordinary Least Squares (OLS)* regression and estimate the *constant* and *variable cost coefficients*. Before going into more depth of the SLM’s selected properties, we may re-write Equation (1) in dilution–cost terms as:(44)$$C\left(m_{ij}^{-1}\right)=\bar{a_{ij}}+b_{ij}\cdot \left(\frac{1}{m_{ij}}\right)$$

In accordance to the generalized formulation of the SP, Equation (44) is suggesting that for each VAC *i* that is recovered by an industry *j*, the total cost of its recovery *C* is proportional to the VAC’s dilution *m_i_*^−1^, multiplied by a *variable cost coefficient b_ij_* and augmented by an independent *constant cost coefficient a_ij_* (capped). Respectively, to the generalized form in Equation (35) and Table 1 with the costs’ taxonomy, parameter *a* concerns all constant-and usually *inflexible*-costs that are indirectly related to the VAC’s recovery volume but have to be paid recurrently at constant time and periodicity, while parameter *b* is a *variable cost intensity* parameter that augments total cost proportionally to the VAC’s dilution level. It is of course obvious that the cost structure of the linear SP reveals the industry’s cost ontology, while it affects significantly its position and share in the market. For instance, from Figure 5b, we conclude that the values of the SP cost parameters demonstrate a clear dominance of the variable cost category and the OPEX class. Cost-ontologically, the unit is *variable cost intensive*, and specifically (by Table 1 classification) is *Variable-OPEX intensive* with only a minimum fraction of constant costs represented by parameter *a* and with the variable costs’ parameter *b* prevailing.

Another significant mathematical SP aspect when VAC dilution is used instead of VAC concentration is the non-constant increases of the fraction 1*/m* across constant decreases of concentration *m* in the denominator. This results to a nonlinear *1/m* growth pattern that follows the general rationale of the *Harmonic Series* similarly to the decay function of *1/n* across linear changes of *n* (with *n*,*m*∈N^+^). Primarily, the use of reverse concentration provides conceptual, graphical and visual SP conveniences as it allows the depiction of the dilution–cost coordinates with increasing values at both axes; beginning from high concentration (low dilution) and, respectively, low recovery cost and ending in lower concentration (higher dilution) with both axes of the variables intersecting at zero (=0). More specifically, any SLM with positive slope depends on the variable cost coefficient that-in turn-defines the *marginal cost* as the *total cost’s differential*. The general condition for the variable cost coefficient to signify that the SP is an SLM is:(45)$$b_{ij}=\frac{\Delta C_{ij}}{\Delta \left(1/m_{ij}\right)}\begin{array}{ccc} =\frac{C_{ij|m_{k}^{-1}}-C_{ij|m_{k-1}^{-1}}}{\left(1/m_{ij|k}\right)-\left(1/m_{ij|k-1}\right)} & & \end{array}$$

Conditions for *C_i_*, *k* and 1/*m_i_* in Equation (45) are ∀*C_i_*∈(0, C_Max_); ∀*k*, (1/*m_i_*)∈(0, (1/*m*)_Max_). Specifically, although the 1/*m_k_* − 1*/m_k_*_−1_ differential alone follows a nonlinear (exponential-type) growth pattern, the ratio *C_k_* − *C_k_*_−1_ follows an identical pattern so that the ratio (*C_k_* − *C_k_*_−1_)/(1*/m_k_* − 1*/m_k_*_−1_) remains always constant with just increasing intervals (as shown in Figure 5b); hence, *linearity is always preserved* for any constant *m* decrease. Alternatively stated, linearity is preserved for any constant “bin” *k* irrespective of its size (*k*∈R^+^). Respectively, the same apply for *per cent* changes [Δ(*C_ij_*)%/Δ(1*/m_ij_*)%] that concern *Cost Elasticity of Dilution* (CED), where the conditions for the SLM are met for constant *CED* = 1.

## 3. Results

In order to validate the theoretical framework of the SP, we utilized data from a very small-scale unit that recovers polyphenols’ (in the role of VAC) from wastewater containing fruit juice residuals. For the unit’s SP model, we identified 5 CAPEX and 5 OPEX cost elements for a range of VAC dilutions and with specific calibration of the unit’s equipment. Indicative cost elements concerned the cost of personnel, the cost purchasing solvents, the cost of electricity etc. In addition, parts of the equipment (e.g., absorption membranes) had to be substituted after a number of VAC recovery iterations. Hence, we considered the completion of these iterations, where a critical part of the equipment had to be substituted, as one *complete cycle of VAC recovery process*. The average (fixed) value of the observed costs from all iterations was adopted as the *indicative VAC cost recovery at every level of VAC dilution*. By repeating the process for a range of VAC dilutions in the wastewater stream, we were able to simulate the unit’s SP. In total, we were able to generate 41 observations/combinations of cost–dilution coordinates for a range of VAC concentrations between 500–100 mg/L of wastewater (or dilutions between 1/500–1/100 mg/L), with a “bin value” (interval) of 10 mg/L of decreasing concentration (increasing dilution) for each different full recovery process.

### 3.1. Empirical Formulation of the Sherwood Plot

A major challenge for the realistic representation of the SP is the distinction between the use of *deterministic data*, which are very useful for establishing an empirical base of the theoretical framework by using fixed cost values and the use of *stochastic data*. The latter addresses to an incomparably more realistic and dynamic depiction of the cost–dilution relationship, also utilizing the full range of the theoretical framework presented in Equations (13)–(43). In our work, we have generated a partial stochastic simulation on the unit’s available data, where we specifically examine the effect of electricity prices’ variability in the market, as an element of total recovery cost, on the unit’s SP.

#### 3.1.1. Sherwood Plot with Deterministic Data

As the unit provided us with fixed values for each of the 10 cost elements that added up to the total cost per dilution level, the measurements provided a perfect linear relationship (R^2^ = 1) between the VAC’s dilution level and the cost for its recovery; hence, resulting to a “deterministic” SP without randomness and deviations around the regression line, as is usually the case of linear regressions and as presented in Figure 1a. Moreover, as we demonstrate in Figure 5b, for each VAC dilution increase by 10 mg/L the total cost for its recovery increases by a factor of 5.220, requiring more man-hours, more solvents, more electricity consumption etc. Generally, the SP’s core assumption (the cost of recovery increases across the increase of dilution due to increasing required effort) is verified even with deterministic data. The incorporation of randomness for more realistic SP depictions is an issue that we discuss in the next section.

As mentioned in Section 2.4.3, the CED is an additional SP metric that concisely describes the per cent response of the VAC’s total cost of recovery towards the per cent change of increase of the VAC’s dilution. Specifically, the CED is calculated as:(46)$$CED=\frac{\Delta C(m_{ij}^{-1})}{\Delta m_{ij}^{-1}}\cdot \frac{m_{ij}^{-1}}{C(m_{ij}^{-1})}$$

As demonstrated in Figure 6a, the CED is quite revealing of the small unit’s cost ontology, verifying the Variable-OPEX classification. For such a small-scale unit this is quite expected due to its upper technical limits of wastewater volume processing for VAC recovery. In turn, this limitation prevents such units from achieving *economies of scale*. Via examining selected complementary metrics, such as CAPEX-OPEX shares in Figure 6b we may confirm the above.

As the observations increase, the CED value asymptotically approaches *1*, confirming the SLM. Regarding CAPEX and OPEX shares, units with potential to achieve economies of scale usually choose to heavily invest on equipment ownership and constant costs with the target of reducing their average cost along the increase of the produced volume. Contrarily, smaller units prefer more flexible schemes for financing their equipment, such as leasing or special renting business models (e.g., pay-by-use).

The separate SP regression on the unit’s CAPEX and OPEX in Figure 7, verifies the dominance of OPEX in the cost structure. Indeed, besides the critical materials, such as membranes and chemicals, even the core infrastructure is financed via leasing agreements, minimizing property management costs and ownership responsibilities. As confirmed by the unit’s managers this is a strategy for simplifying financial management by reducing it to periodical rent-type payments until the expiration of a pre-agreed time, which may or may not coincide to the equipment’s technical life. This practice is frequently adopted by very small-medium enterprises (VSMEs), classified as technology leasing. In overall, we may concentrate the unit’s cost parameter values for each separate regression (CAPEX, OPEX, TOTEX) in Table 2:

The above CAPEX and OPEX coefficients represent the shares shown in Figure 6b, where also their sum is equal to the TOTEX coefficients. The deviation of the constant TOTEX coefficient from the sum of constant CAPEX and OPEX coefficients can be considered negligible and is even completely eliminated for the expansion of the sample to VAC concentration levels as low as 10 mg/L. In addition, how the 5 CAPEX and 5 OPEX cost elements are specifically distributed between constant and variable cost categories is beyond the scope of our work and is left for a future publication.

#### 3.1.2. Sherwood Plot with Stochastic Data

As the above deterministic approach was a useful experiment and starting point for the verification of the SP’s core assumption, we needed to expand our measurements into a simulation and add the effect of *variability* in the market prices of the unit’s cost elements. Specifically, as an industry purchases critical supplies (e.g., materials) and services (e.g., maintenance) periodically, the change of (international) market prices fundamentally affects its revenue and profit predictions and -hence- its annual balance sheets.

For simplicity, we chose *electricity prices* as only one of the ten cost elements to consider that prices are now fluctuating. Equivalently, it is as if we had no prior information from the unit on its electricity cost and had to estimate it -as part of its SP- via external data. All of the other nine cost elements were still considered non-variable. For our case study, we examined the effect of variability in wholesale prices of electricity traded in the Hellenic electricity market as crucial element of the unit’s VAC recovery variable OPEX.

In Figure 8a, we present the patterns of the *System Marginal Price* (SMP) and the *Clearances Marginal Price* (CMP) recorded by the *Independent Power Transmission Operator* (IPTO) [61], for a period between November 2019 to October 2020 with daily frequency (365 observations in total). The SMP is the *ex ante* (day-ahead) price of electricity based on the day-ahead *predictions* of the electricity load demand, while the CMP is the *ex-post* (day-after) day of electricity based on the *observed* electricity load demand. Deviations between predicted and actual load -also causing SMP and CMP deviations- occur always. As it is demonstrated in Figure 8b there is a strong correlation between the SMP and the CMP; minimizing the day-ahead and day-after deviations. However, in any case, calibrations in the grid will inevitably re-configure the load shares per energy source; hence their final wholesale and retail prices as well, making the CMP as the most suitable metric to address our case study’s final electricity cost. The stochastic model for the electricity cost of our case study is the following:(47)$$C(E_{\rho })=f_{N}[C(E_{\rho MAX});\sigma ]\cdot \left[1+\frac{\left(\frac{\Delta C(E_{\rho })}{\Delta \rho }\right)\cdot \left(C(E_{\rho MAX})-C(E_{\rho })\right)}{k}\right]$$

For a VAC concentration range of 500–100 mg/L, the electricity cost in the unit ranges between 0.052–0.26 € that -as share- remains relatively constant, accounting for 0.48% of the total recovery cost. Starting from a maximum concentration *ρ* of 500 mg/L we estimate an average electricity cost increase by 0.0052 € or a 10% electricity cost increase for every 10 mg/L of VAC concentration decrease. The probability density of the CMP as the basic electricity cost *f_N_[C(E_ρ__MAX_)]* follows a normal distribution with a convergent mean value of 0.052 €/kWh for *ρ_MAX_* = 500 mg/L and standard deviation of 0.012. Although the CMP concerns wholesale prices that are accessible by larger scale industries we assume for simplicity that they are accessible by even this very small unit.

In overall, the CMP’s moments (mean, standard deviation, skewness and kurtosis) suggest normality as presented in Figure 9a. After their incorporation to the model, we introduced the constant coefficient of the electricity cost increase of 0.0052 € per 10 mg/L of VAC concentration decrease, normalized by the bin value *k*. This coefficient is a cost multiplier of the basic electricity cost of 0.052 €/kWh at *ρ_MAX_* and increases the electricity cost across the VAC’s dilution increase. Whatever uncertainty may be affecting the basic electricity cost per kWh, the trend depends exclusively on the dilution level.

The next step was to perform a Monte-Carlo simulation for the cost of electricity by generating 1000 different SPs by the CMP’s moments. Each of these iterations generated a separate SP. We then used the range of these iterations to regress the unit’s new optimal SP by the OLS method. With this approach the cost or electricity was found to constantly account for 0.48% of the TOTEX at each VAC dilution level for all the 500–100 mg/L range. As demonstrated in Figure 9b the points of the new SP are slightly lower than the regression with deterministic data and the total cost coefficient increase for every increase of dilution (by 10 mg/L) was reduced to 5215 from 5220 -or by 0.1%- while the constant cost coefficient was reduced from 0.347 to 0.290 -or by 16.43%. Although the unit’s electricity cost accounts for only a small fraction of the TOTEX, the sensitivity of cost coefficients in the stochastic SP varies significantly. Specifically, for the change of the base electricity price by 52.72% (from 0.11 to 0.052 €/kWh) the variable cost coefficient is extremely insensitive, diminishing by only 0.1% (527.2 times less), while the constant cost coefficient more sensitive, diminishing by 16.43% (3.2 times less).

It is of course obvious that multiple correlations and interdependencies and at various time lags between the unit’s 10 cost factors take place. Had we repeated this rationale for the rest of the 9 remaining cost elements, the unit’s SP would probably be even more variable. However, the incorporation of randomness in even one of the 10 cost elements verified the fundamental SP argument. Due to its complexity and analytic requirements, performing the analysis for all cost elements escapes from the scope of our current work and is left for a future publication.

### 3.2. The Sherwood Plot and Circular Economy Finance (CEF)

Empirically, it can be said that while all industries incorporate a level of market price uncertainty on their projected revenues and costs, at a secondary level high market price variability and/or volatility is usually dealt with forward contracts that “lock” purchase prices for both counterparties; buyers and suppliers. From a cost-ontological perspective an industry may choose to shift its cost factors from constant to variable and vice versa. The most frequent shift though is towards constant costs to hedge fundamental macroeconomic risks (e.g., the discount rate) that may derive from extremely volatile markets. Price volatility threatens an industry’s liquidity, credit rating, shareholders’ trust and eventually its stock market value. The main financial instruments for dealing with the variability of sensitive supply chains mainly concern derivatives contracts, such as options and futures. The major feature of derivatives contracts is that the total risk is not mitigated but mainly re-allocated towards the counterparty of the (derivative contract) agreement that is willing to take it.

For instance, an industry may need to secure the supply of its critical materials (e.g., solvents, chemicals) to be used as production factors in the recovery of a target VAC, so that it may report to its shareholders its balance sheet state with a higher certainty. To achieve that, wishing to avoid the event of an extreme volatility in international markets it may sign a *futures* contract with a supplier in order to receive at a future time (e.g., 3, 6, 9 months from now) the materials at a fixed price that is determined *now* (which is 3, 6 or 9 months prior to the physical delivery). With such a contract, the risk is not neutralized or mitigated but just *transferred* to the supplier who accepts to take the risk of lower profits via *pre-selling* at a current lower *fixed* (thus *certain*) price than at a *potentially higher* at the time of physical delivery. In fact, derivatives contracts can partially work as substitutes of risk mitigation for price stabilization and cost accounting predictability. 

Derivatives contracts can take the most exotic forms; applying to many processes and to a plethora of industries, such as energy supply or agriculture that are highly depended on weather phenomena [27,62]. In the SP context, such contracts can be tailored to apply to industrial ecosystems, with underlying indices concerning the concentration prerequisites of a target VAC within a wastewater matrix to be accepted for further processing and recovery, or of even the composition of the wastewater matrix itself.

### 3.3. Market Structure and the Sherwood Plot

After identifying the dilution–cost coordinates and functions of industries, the next significant step is to understand how a typical VAC recovery market would form and further operate from the composition of at least two (*j ≥ 2*) industries. This part of our work is an introduction to the first important aspect of the SP’s microeconomic extensions. Thus, this section is separated in three major parts: (a) the process and necessary conditions of a VAC’s recovery *market formation* based on the SPs of at least two industries operating in the market with the same objective, (b) the *optimal allocation* of the VAC’s recovery by its dilution level when received and (c) a basic set of *Market Structure Diagnostics* (MSDs) to identify the tendency of the market to either concentrate in favor of specific industries or to operate in a highly decentralized and competitive manner.

#### 3.3.1. Market Formation Process

For simplicity, we assume the existence of only two (2) industries operating in the recovery of the same VAC. The industries are assumed to have different SP parameters that define their individual constant and variable cost intensities. Other SP forms, even the most theoretical ones with discontinuous or even with a constantly negative gradient, implying extremely large economies of scale cannot be excluded. Additionally, in cases where many industries operate, the market SP could have a quite complex form. In any case though, the level of complexity does not affect negatively the fundamental validity of the SP, both for individual industries and for the market structure.

To compose a VAC’s recovery market we need to define some fundamental characteristics of the two individual industries as its elements: *Industry A* and *Industry B*. We have already assumed a difference in their parameters, where specifically *a_iA_ < a_iB_* and *b_iA_ > b_iB_*, according to Equation (44). We also assume that both industries have linear SPs. Additionally, we assume that both industries have a common *technical limit* and *operational range*; meaning that there is an upper VAC dilution level after which none of the industries can technically achieve its recovery. Both industries can operate (to recover the VAC) in the range between that technical limit and zero VAC dilution, meaning that both industries have exactly the same technical capabilities. With these assumptions, Figure 10 depicts the formation of the market’s VAC recovery SP as a composition of the two individual SPs.

Based on the above model, as each industry is more efficient at specific sub-domains of VAC dilution, the market will allocate respectively industry shares by *operational dominance*; that being where the VAC’s recovery is achieved at the minimum cost. In our example, *Industry A* as *variable cost intensive* dominates in a range of lower *VAC Dilution Index* (DI) values (*DI*∈[0, 25)), while *Industry B* as *constant cost intensive* dominates in the domain of higher dilutions (*DI*∈(25, 50]). The SPs of the two industries intersect for a VAC DI value of 25, at the only dilution level where both industries can recover the VAC at the same cost. This is the *operational shift point*, as across an ascending or descending sorting of the VAC DI, before or after that point, another industry starts to become more cost efficient and, hence, begins to dominate in the VAC’s recovery. In our example, the operational shift concerns the transition of the VAC’s recovery from industry *A* to *B* that utilizes its potential *economies of scale* (as it is usually the case with constant cost intensive industries). However, the total number of operational shift points is relative; depending on the number of industries operating in the *maximum technical range*.

#### 3.3.2. Market Allocation and Optimization

Even these simple market formation dynamics are able to generate numerous varieties of market SPs. In any case, the allocation of the VAC’s recovery via the superposition of SPs of individual industries has a specific economic content. Primarily, we observe significant cost structure diversification in industries *A* and *B*. Specifically, for low VAC dilutions, Industry *A* is more cost efficient than *B*; however after the intersection point (for VAC DI = 25), Industry *B* becomes more cost efficient in higher dilutions. In turn, Industry *B* is generally less depended on variable costs due to low value of parameter *b_iB_* and more depended on constant costs due to high value of parameter *a_iB_*, as the increasing VAC dilution level does not affect significantly its total cost increase both in absolute and in (%) terms.

Additionally, although in our example both industries share the same technical range, in reality some of the industries could stop being technically operational (not being able to recover the VAC) at a much lower VAC dilution level; even for a VAC dilution value within their theoretical dominance (that would be for *Industry A* to stop being technically operational at a DI value <25). In that case, assuming that *Industry B* maintains its technical limit equal to the market’s maximum (in our example for a DI value = 50), it would fill this market gap by taking over the technically non-operational part of *Industry A*. In such a case, the market SP would have a discontinuous vertical “jump” from the SP of *Industry A* until it meets the SP of *Industry B*. It is more than obvious that this case is only indicative and that a variety of many more possible combinations could be generated.

The *Market’s Technical Limit* (*TL^M^*) defining the technical range for *all* industries that operate in a specific *i* VAC recovery, irrespective of whether all industries meet that technical limit or operate only within a sub-range- is:(48)$$TL^{M}(m_{i}^{-1})=Max\{Max[TL(m_{ij}^{-1})];\overset{n}{\underset{j=1}{\cup }}TL(m_{ij}^{-1})\},\forall j,m_{i}$$

The condition in Equation (48) sets the *market’s technical limit* as either of the union of all technical limits of industries operating in the market or of the industry with the highest operational range for the case where *TL*(*m_i_*_1_^−1^)⋃;...;⋃*TL*(*m_in_*^−1^)⊆*Max*[*TL*(m*_ij_*^−1^). Hence, according to these conditions we may write:(49)$$\overset{n}{\underset{j=1}{\cup }}TL(m_{ij}^{-1})=TL(m_{i1}^{-1})\cup TL(m_{i2}^{-1})\cup ;\ldots ;\cup TL(m_{in}^{-1}),\forall j,m_{i}$$

Additionally, with more industries operating within any continuous technical range, the number of the market’s *operational shift points* will be respectively higher. In general, the number of operational shift points *u^M^* for a number of *j* industries operating in the recovery of a specific *i* VAC is:(50)$$u^{M}(m_{i})=\left(\sum j\right)-1,\forall j,m_{i}$$

According to the cost conditions described above, considering that all industries operate in the same technical range, the *minimum cost* for the recovery of VAC *i* at dilution *d* is defined by Equation (51) as:(51)$$C^{M}(m_{id}^{-1})=Min[C(m_{id|j}^{-1})]=Min[C(m_{id|1}^{-1});C(m_{id|2}^{-1});\ldots ;C(m_{id|n}^{-1})],\forall d,j,m_{i}$$

For an integrated and parsimonious depiction of the necessary conditions for a VAC’s recovery market formation, described by Equations (48)–(51) as the optimal allocation of specific *i* VAC’s recovery across a *j* number of industries by the criterion of *minimum cost at every dilution level d* and considering the technical constraints *TL* of each single industry *j*, we may write:(52)$$C^{M}(m_{id}^{-1})s.t.TL^{M}(m_{i}^{-1})=Min\{C(m_{id|j}^{-1})|[TL^{M}(m_{ij}^{-1})]\},\forall d,j,m_{i}$$

In short, Equation (52) states that the market will choose for the VAC’s recovery at every dilution level the industry with the minimum cost, but only among those industries that can technically operate at this specific dilution level. Provided that the conditions of Equations (48)–(52) are met, the superposition of the industries’ SPs actually reveals the sequence of dilution–cost coordinates formed by the industries, with the market allocating efficiently the recovery of the VAC by assigning it—for every dilution level *d*—to the industry with both the technical capacity and at the lowest cost to implement the recovery. To assess how economically efficient is the market across its *TL^M^* we write:(53)$$[C^{M}(m_{i}^{-1})]s.t.TL^{M}(m_{i}^{-1})=Max\{[C(m_{ij}^{-1})]|[TL(m_{ij}^{-1})]\},\forall ,j,m_{i}$$

Equation (53) simply suggests that the market’s VAC recovery total cost range is the maximum cost of VAC recovery across the union of the distinct technical limits of industries operating in it, as described in Equations (48) and (49). Here, the main difference with the conditions for the technical limits, is that contrarily to the TL^M^ -where even the case of *n* − 1 industries’ TL unions being a subset of industry’s *n* TL does not necessarily exclude them from operating in the VAC’s recovery market- for the cost range C^M^, if the *n* industry also recovers the VAC for the whole TL^M^ at the lowest cost at every dilution level *d*, it will definitely exclude every other industry and will -subsequently- monopolize the market.

#### 3.3.3. Market Structure Diagnostics (MSDs)

A critical structural aspect of markets’ efficient operation concerns their *concentration*. Utilizing our previous arguments on cost ontologies, the primary objective of industries will be to collect and monitor a coherent set of representative *Market Structure Diagnostics* (MSDs) on the allocation of the full range of the *dilution*–*cost* relation (from zero VAC dilution to the upper technical limit of its recovery) in market *segments* where each industry dominates. Comparing the segments of both the recovery and the cost range, we may extrapolate significant conclusions on the market’s structure and inequality of the industries comprising it.

Specifically, the measurement of industrial ecosystems’ performance has been identified from very early as a central issue to differentiate them from mainstream sustainability solutions [63]. The idea of industrial ecosystems began rather optimistically, promoting a model of fundamental economic re-structuring via the incorporation of ecosystem limitations and carrying capacities [64] and continued growing at the same spirit [65], as at its initial stages the field was uncharted and the capacity for profitable wastes’ reuse was huge. However, as many industrial symbiosis clusters started to meet their maximum technical or biophysical capacities, with the contribution of conventional technology offering only a marginal improvement, a founded skepticism emerged. Concerns were expressed on the risks of whether societies truly made a difference or if they just invented a different conceptual framework to continue justifying the life extension of existing industrial paradigms instead of substituting them [66].

In relation to the above, we develop four (4) MSDs for the integrated assessment of the market’s structure as complementary to existing market concentration indices. We begin from the *Herfindahl*–*Hirschman Index* (HHI) as a quick and widely used market concentration index, based on the squared company market shares (*s*):(54)$$HHI_{ij}=\sum_{i|j=1}^{n}s_{ij}^{2}$$

The *HHI*∈[1/n, 1], although a simple, quick and representative overall market concentration index—especially for its upper and lower limit values—provides a more obscure picture for its intermediate values, while it does not take into consideration other aspects, such as the market’s hierarchy (for markets with imperfect competition), as well as the allocation of control on the market’s price range. For these reasons, we begin to build our MSDs with the *Operational Prevalence Index* (OPI), depicting the share of a *j* industry’s control on the operational range of the *i* VAC’s recovery. Simply stated, the *OPI* shows for which part of the *i* VAC’s dilutions total range (from zero to the upper technical limit) an industry *j* prevails against others with a minimum cost:(55)$$OPI_{ij}=\frac{\overset{TL^{M}}{\underset{m_{iMIN}^{-1}}{\cup }}\left\{m_{ij}^{-1}|Min[C(m_{ij}^{-1})]|[TL^{M}(m_{ij}^{-1})]\right\}}{TL^{M}(m_{i}^{-1})}$$

The *OPI*’s values range from 0→1 {*OPI*∈[0, 1]}. It is expected empirically that in the majority of cases the operational shares will be distributed unequally between industries. However, even in the special case where the operational shares are equally distributed, a question remains on what part of the cost range is controlled by each industry. Respectively to the *OPI*, we develop the *Cost Prevalence Index* (CPI), which—following the same rationale—measures what fragment of the *i* VAC’s recovery costs total range an industry *j* is able to control due to its operational prevalence:(56)$$CPI_{ij}=\frac{\overset{TL^{M}}{\underset{m_{iMIN}^{-1}}{\cup }}Min\{C(m_{ij}^{-1})|[TL^{M}(m_{ij}^{-1})]\}}{Max[C^{M}(m_{i}^{-1})|TL^{M}(m_{i}^{-1})]}$$

Respectively, to the *OPI*, the *CPI* values range from 0→1 {*CPI*∈[0, 1]}. For an SP analysis, the cost range control is a very significant aspect of the VAC’s recovery market, as an industry may have a higher ability to control the cost range than the operational range. Specifically, that means that although an industry may be more efficient in recovering the VAC for only a small fraction of the *TL^M^*, it may be disproportionally more able to control the VAC’s recovery cost range. Simply stated, in such cases the VAC’s recovery market tends to operate more through the control of the recovery costs by a small fraction of industries rather than the ability of the industries to recover the VAC competitively at lower costs. In relation to the *HHI—*as a measure closer to the *OPI—*the *CPI* provides a picture on industry’s power to control market costs. A high value of the CPI is a vital diagnostic of *imperfect competition* and *market concentration*. In this context, the constant and variable cost parameters also become *normative* metrics in addition to those that are positive.

We may integrate the above concepts and argumentation by introducing the *Market Prevalence Index* (MPI) as the ratio between the *OPI* and *CPI* as:(57)$$MPI_{ij}=\frac{OPI_{ij}}{CPI_{ij}}=\frac{\frac{\overset{TL^{M}}{\underset{m_{iMIN}^{-1}}{\cup }}Min\{m_{ij}^{-1}|C(m_{ij}^{-1})|[TL^{M}(m_{ij}^{-1})]\}}{TL^{M}(m_{i}^{-1})}}{\frac{\overset{TL^{M}}{\underset{m_{iMIN}^{-1}}{\cup }}Min\{C(m_{ij}^{-1})|[TL^{M}(m_{ij}^{-1})]\}}{Max[C^{M}(m_{i}^{-1})|TL^{M}(m_{i}^{-1})]}}$$

In turn, the *MPI* provides a concise view of an industry’s *prevalence intensity*. In simple words it assesses whether an industry relies more on its ability to operate more competitively than the others in terms of *TL^M^* share or if it relies more on its ability to control a large fragment of the recovery cost range. The MPI’s range of values is from 0→+∞ [*MPI*∈(0, +∞)]. Its extreme values (MPI = 0, +∞) are only theoretical (thus the “open” set in contrast to the OPI and CPI), as they would signify that an industry is either not participating in the market at all (for *MPI* = 0) or that it offers the VAC’s recovery at zero cost as it controls no segment of the VAC’s recovery cost range (*for MPI = +∞*). Within the above context, we can visualize the cost ontology elements of each individual industry and the VAC recovery market to deduct significant conclusions on their current technical and economic efficiency. In the case of an aggregate view, if each industry forming the market has different constant and variable cost parameters, the market’s overall SP acquires a nonlinear shape, with its regression parameters providing its overall economic efficiency. A graphical analysis of Equations (55) and (56) on the industries’ operational and cost features, as well as of the unified VAC recovery market formed by the integration of the two industries’ features, is presented in Figure 11a,b below:

For the three (3) possible MPI values in between, their economic meaning is the following: (a) for *0 < MPI <* 1, the industry relies more on the control of the recovery cost range than of the operational range; (b) for *MPI =* 1, the industry is “fair” as it controls the recovery cost range proportionally to its operational ability. At this point it is of imperative importance to denote an additional mathematical feature of the MPI concerning its rare conditions for acquiring a value of (=1). These conditions are: (1) all industries must have exactly the same VAC recovery range (hence equal OPI); (2) the operational ranges of industries have to form a *perfect sequence*, meaning that where the one industry stops operating the next begins so that the *TL^M^* is formed by a *perfect union* of its separate segments; and (3) all industries have the *same* parameter value of variable cost coefficient. In reality, as these conditions are extremely difficult to be met, this case comprises a very good approximation of the observed industrial organization and imperfect competition (oligopoly) phenomena. The last case is (c) for *MPI >* 1, the examined industry operated in a more competitive context as it controls a smaller fraction of the VAC’s recovery cost range than its control on the operational range.

Moreover, the *OPI* and *CPI* as constituents of the *MPI* can serve as approximates for the *Shannon MaxEnt*, for the special case where the sector consists exclusively of *n* (*n*∈N^+^) “fair” industries (with *OPI* = *CPI* = 1*/n*∀*n*; *MPI* = 1), so that the *Shannon Entropy* is maximized and is proportional to the number of industries (=*n*) forming the market.
(58)$$H{\left(n\right)}_{MAX}=-\ln\left(\frac{1}{n}\right)\propto OPI_{ij}=CPI_{ij}=\frac{1}{n},\forall j\in N^{+}$$

Finally, the *MPI—*along with the *OPI* and the *CPI* as its constituents*—*can be utilized with ordered data (ascending *Min*→*Max*) to depict a *Lorenz Curve* as a metric of *recovery cost inequality* in relation to the shares of the operational range. A typical Lorenz graph depicts in ascending order “what percentage of the *Y* variable is controlled by an ascending *X* percentage of elements”. Regarding the SP on a VAC’s recovery, the postulation would be “what percentage of the recovery cost range is controlled by the *X* percentage of the operational range”. From that point and after, the principles applying to a typical Lorenz Curve on allocation inequality, apply for the MPI as well.

## 4. Discussion and Extensions

In the previous sections we developed a theoretical and quantitative argumentation on how a SP can depict a VAC dilution-cost relationship and constitute a solid framework for identifying a single industry’s *cost ontology*. The identification of the cost ontology can further relate to an industry’s cost strategy on optimal VAC recovery. However, within an *industrial symbiosis cluster*, the single industry constitutes only an element, of which the importance is defined by the related MSDs. Moreover, even an industrial symbiosis cluster is just a *cell* of an industrial ecosystem -among various diversified cells- that in overall aim at maximizing the conservation of *energy*, *mass* and *information* inputs from natural capital utilization [60] in terms of *quantity*, *quality* and *residence times*. Eventually, the above *biomimetic* concept of the *Circular Economy* -in contrast to the *Linear Economy*- is consistent to these principles.

### 4.1. Market Ontologies

Extending the above rationale, the next discussion would concern how a symbiotic cluster operates in relation to its separate elements. Do individual industries operate differently alone in the competitive market than as part of a symbiotic cluster? Is it feasible to keep the same level of autonomy while operating in a synergistic business model? Do “local optimizations” of individual industries oppose to “global optimizations” of the cluster? Would a single industry even have an incentive to participate in a symbiotic cluster? And what metrics would it adopt to make this decision? These questions on operational optimizations of symbiotic clusters and their large-scale manifestation, beyond an ideal construct are indicative in order to provide a typology. Primary attempts to substantiate philosophically a theory of *industrial ecology ontologies* [67] turned towards the need of adopting a new conceptual framework, based on mathematical rules for including complexity, mutual exclusivity and structural autonomy.

To this approach we could add that both from philosophical and computational point of view, an ontology is an integrated, self-sufficient—from the microscopic level to the macroscopic and vice versa—set of properties that define a deductively consistent and topologically independent reproducible mathematical construct. Ontological self-sufficiency consists in the feature that the construct consists of all the necessary properties to be considered an exactly complete entity. The mathematical properties described across Equations (1)–(43) constitute the ontological elements of single industries that engage in VAC recovery, while Equations (48)–(58) comprise ontological elements of two or more industries forming a VAC recovery market entity. The industrial symbiosis market is a system with an ontological expansion of single industries as its distinct components [68]. Besides their pure mathematical and computational conveniences, these sets describe the properties defining the generalized model of the SP (as the empirical plot by itself cannot be an ontology) with the ability to incorporate smaller ontologies within the larger; hence, making the ontology scale-free verifiable. An example of this property is that the dominance of constant cost or variable cost ontologies can be found both in the individual industry and the market entity.

### 4.2. Research Extensions of the Sherwood Plot

The SP constitutes a remarkably flexible and functional framework, applicable to numerous species of *Ricardian resources*, which are resources of which the *total quantity is qualitatively diversified*. In more detail, this means that the total quantity of a natural resource (whether virgin or VAC under recovery) is distributed in various unequal *quality levels*. The SP depicts qualitative diversification via the *cost of recovery to dilution* relation. Whether we focus on virgin or secondary resources, *resource quality* is directly related to *molecular purity*. At the industrial processing level, molecular purity can further be defined as *a material’s desirable molecular sequence based on an adopted standard in relation to the objective that the molecular sequence is expected to achieve*. Any deviation from this optimal molecular sequence signifies lower quality as it does not optimally satisfy its specific objective. Even in cases of higher purity, such as demineralized water, what is ideal for various industrial uses is not ideal for drinking water.

In turn, molecular purity (thus, resource quality) is directly related to its higher concentrations within a matrix of materials; whether we adopt *micro-statistical* quality evaluation, such as concentrations for virgin mineral ores in mining, VAC concentrations in wastewater, *Gibbs Free Energy* for chemical reactions, enthalpy for geothermal energy fields, olive oil acidity degrees (also impacting on polyphenols’ extraction costs) or *macroscopic* quality evaluation, such as *Energy Return on Energy Invested* (EROEI), *exergy* surplus for the increase of an economic system’s structural complexity [60], adaptability and robustness towards external distortions [69]. In addition, the SP is compatible to more abstract quality evaluation criteria, such as unified (non-vector) water quality indices, discrete entropy states of molecular systems [70], as well as any well-defined structured index that evaluates natural capital quality in terms of its deviation from a reference state.

At a more technical econometric level, a pillar of the SP’s future general utilization concerns the identification of high impact cost factors in process engineering sequences. After the identification of these factors, a next step would concern the decision of how to cope with them. That consists in choosing between (i) *defensive* strategies via derivatives (futures, options) for the supply of critical materials or (ii) *aggressive* strategies via investment in new technologies for triggering a *structural change* in the sector and acquiring higher market shares. However, this option always comes at a risk, as R&D programs are incorporating significant uncertainty in the result [71], while respective responses from competitive industries always pose the risk of cancelling out the initial effect.

Finally, it is important to denote that industrial symbiosis clusters are not the ideal of competitive industries, as by definition they aim at synergies and types of ad hoc corporate agreements [72]; such as *Chemical Leasing* (ChL) contracts [73] that aim at maximizing VAC recovery, minimization of environmental costs and the optimal allocation of profits. A more technical facet of the econometric analysis would also concern issues of correlation (linear or nonlinear) between two or more VACs under recovery. In particular, in our case study, having assumed a binary mix that consists of only water and the VAC, their *complementarity* is straightforward since the VAC’s recovery and retrieval from the mix leads by default to the purification of the residual water. The generalization of this rather simple case to more complex mixes would be a cutting-edge research extension for industrial ecosystems, as correlated or “Granger causal” [74] recoveries would imply lower costs for synergistic business models.

## 5. Conclusions

Our work investigated the properties of the *Sherwood Plot*. Originally, the Sherwood Plot concerned the relation between a target material’s dilution in a waste matrix and its required market price in order for its recovery to be considered economical in comparison to its extraction from a virgin ore. After the in-depth examination of its various physical an economic foundations, we conclude with the following findings: (1) we developed an argumentation on the Sherwood Plot’s conceptual and mathematical re-postulation to depict the relation between a target material’s dilution and its cost of recovery as a more straightforward model statement with higher operational convenience; (2) we then developed a parsimonious (with least possible assumptions) and very analytical mathematical framework for the Sherwood Plot’s generalization, with coherence from the microeconomic to the macroeconomic level; for two or more industries forming a market we examined the emergent properties of a target material’s recovery market. In addition, the above context’s structure allowed us (3) to examine conceptualizations of higher abstraction, such as the cost (for the single industry) and market ontologies.

Having developed the theoretical mathematical framework (4) we empirically examined the case of a very small pilot unit of polyphenols’ recovery from industrial wastewater after natural fruit juice processing. Through the case study, we presented many significant metrics that are important to the theoretical framework developed before (i.e., the cost ontology), as well as to the identification of significant standard economic metrics, such as the elasticity of the total cost of recovery and the constant and variable cost shares across the increase of the target material’s dilution. Concerning the empirical study (5), we also made the distinction between deterministic and the stochastic approaches (as closer to real-world cases) with an example of variable electricity prices, using daily data from the *IPTO* for a 12-month period. Via our stochastic simulations, we depicted how the cost variability of production factors may affect an industry’s Sherwood Plot. Getting deeper into the structure of recovery markets, (6) we developed a set of *Market Structure Diagnostics* (MSDs) that classify market ontologies. Finally, in the end of our integrated work, (7) we discussed the Sherwood Plot’s various cutting-edge research extensions.

## Figures and Tables

**Figure 1 entropy-25-00004-f001:**
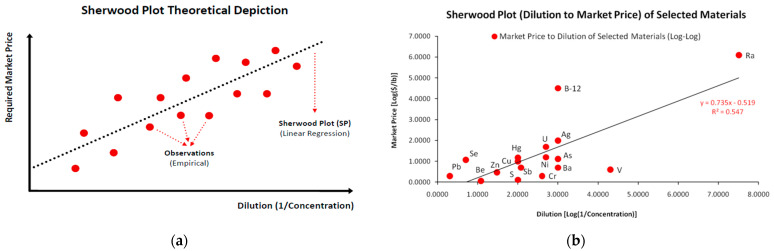
(**a**) Schematic (theoretical) depiction of the Sherwood Plot on the original relationship between the *dilution* (defined as the reverse concentration ratio, equal to *1/Concentration*) of an identified VAC in a waste matrix and its *required market price* in order for its recovery to be considered economical for a sequence of dilution levels; (**b**) Restructuring an empirical linear Sherwood Plot with data from the National Academy of Engineering [13] for selected elements and materials along with its *R*^2^ value (=0.547). The reconstruction is consistent with the theoretical model, depicting the relationship between materials’ dilution [as *Log*_2_*(1/Concentration)*] in a waste matrix and their required market price [as *Log*_2_*($/lb)*] for their recovery to be considered economical.

**Figure 2 entropy-25-00004-f002:**
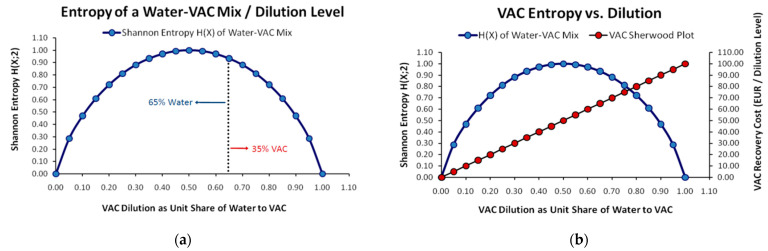
(**a**) Schematic representation of the *Shannon Entropy H(X)* non-monotonic graph of an indicative mix containing only *Water* and the *VAC* at concentration levels of 65% and 35%, respectively. Alternatively, we may consider a *statistical probability* of finding one VAC molecule among the total number of molecules (the sum of VAC and Water molecules) forming the mix, equal to 35%; (**b**) The *Shannon Entropy H(X)* non-monotonic graph vs. a typical SP monotonic graph (here linear) in superposition, depicting the increasing recovery cost of a VAC as it becomes statistically increasingly scarce within the mix.

**Figure 3 entropy-25-00004-f003:**
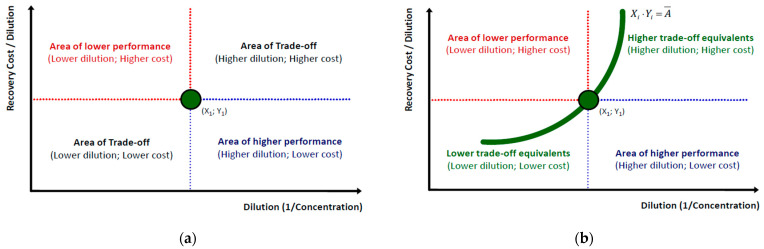
(**a**) From *any* initial random coordinate (*X*_1_,*Y*_1_) of VAC dilution (*X*_1_) and recovery cost (*Y*_1_), there are four (4) compartments (here symmetric for simplicity) automatically formed, having the coordinate as their centre. Specifically, the *lower right* area signifies all combinations of *Pareto Improvement* with lower recovery costs even at higher dilutions; the *upper left* area signifies all combinations of *Pareto Deterioration* with higher recovery costs even at lower dilutions. In the two (2) remaining areas -lower left and upper right- a trade-off takes place with higher (lower) dilutions and lower (higher) recovery costs respectively. These are the areas of uncertainty (with no definitive answer on whether a Pareto improvement or deterioration takes place), in which a SP exists, crossing through specific coordinate combinations; (**b**) Within the trade-off areas, an *Indifference Curve* depicting all *equivalent* to the initial (*X*_1_,*Y*_1_) coordinates is formed. For maintaining *constant* the *per unit cost* of a VAC’s recovery (*C_i_/m_i_*), the *total cost C_i_* will have to increase across an increase in dilution *1/m_i_* and respectively decrease when the dilution decreases. The indifference curve is a *unique* chart of *infinite equivalent solutions* to any *initial coordinate*.

**Figure 4 entropy-25-00004-f004:**
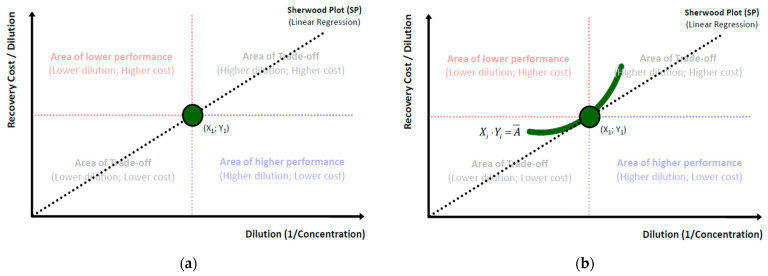
The microeconomic formation of the SP; (**a**) The SP crosses through the compartments of indefinite Pareto improvement or deterioration (from lower left to upper right); (**b**) A linear SP is also the tangent of an infinite set of (*Xi*,*Yi*) coordinates, from which derive infinite combinations of equivalent solutions with constant per unit cost of recovery. The charts’ coordinates that also belong to the SP (as their mutual tangent) form the optimal dilution-cost set of coordinates.

**Figure 5 entropy-25-00004-f005:**
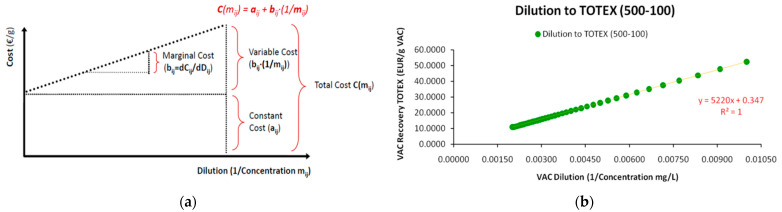
(**a**) Schematic depiction of a linear SP’s *main* and *derivative* cost categories are analyzed; *constant*, *variable*, *total* and *marginal*; (**b**) Empirical SP with *Total Expenditure* (TOTEX) utilizing real data from a small-scale unit for the recovery of a polyphenol in the role of the VAC from wastewater and for a range of VAC concentration between 500–100 mg/L and cost coefficient values *a_i_* = 0.347 and *b_i_* = 5.220.

**Figure 6 entropy-25-00004-f006:**
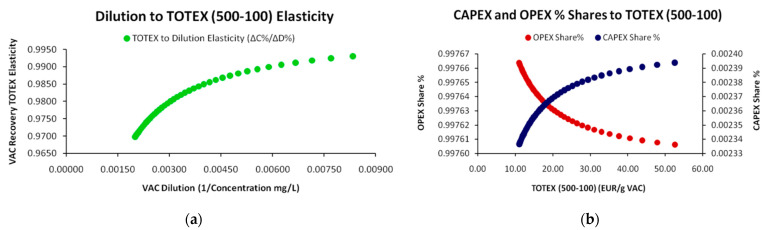
(**a**) Elasticity of TOTEX per level of VAC dilution with a value asymptotically approaching *1*, confirming the SLM; (**b**) Change of CAPEX and OPEX shares to TOTEX across the increase of VAC dilution levels. The OPEX is overwhelmingly dominant in the unit’s cost structure with an always higher share than 99.7% in the SP while the share of CAPEX is limited to a maximum of 0.0024%.

**Figure 7 entropy-25-00004-f007:**
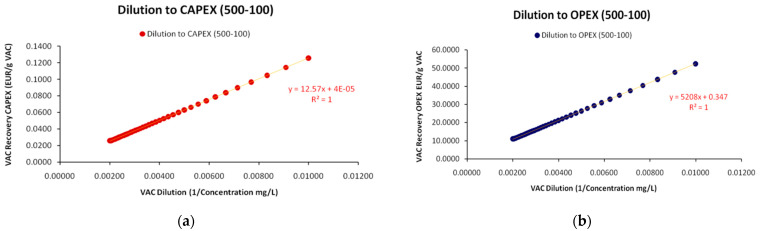
Separate SPs for: (**a**) CAPEX and (**b**) OPEX. It is observed that CAPEX, although increasing along the increase in the VAC’s dilution, it comprises a minor fraction of the TOTEX as both the *constant* and *variable cost* parameter values (*a* = 0.0269; *b* = 12.57) are extremely low in relation to the respective OPEX ones (*a* = 0.347; *b* = 5.208).

**Figure 8 entropy-25-00004-f008:**
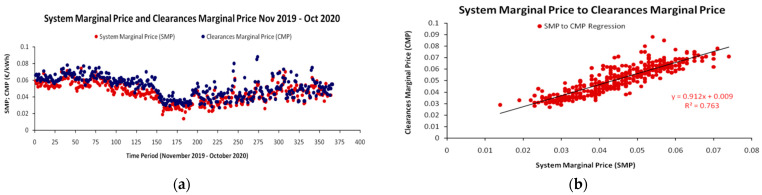
Testing the effect of variability of electricity market prices on the TOTEX and SP of the unit; (**a**) The variable price of electricity in Greece was examined for a period between November 2019 to October 2020 with daily frequency (365 points) for both the SMP and the CMP; (**b**) Although the SMP is frequently used as reference price, the final and corrected market price is the CMP. In any case, empirically there is a strong SMP-CMP correlation observed.

**Figure 9 entropy-25-00004-f009:**
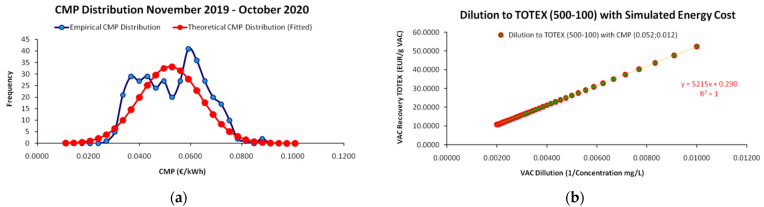
Empirical and theoretical distribution of the electricity CMP: (**a**) The calculation of its moments across the examined year suggests normality; (**b**) The distribution’s standard deviation was used to simulate a variable CMP, affecting the unit’s TOTEX at each VAC dilution level. The simulated *constant* and *variable cost* parameter values (*a_i_* = 0.290; *b_i_* = 5.215) account for a 16.43% and 0.1% change respectively in relation to the estimated ones with deterministic data, making the constant cost more sensitive.

**Figure 10 entropy-25-00004-f010:**
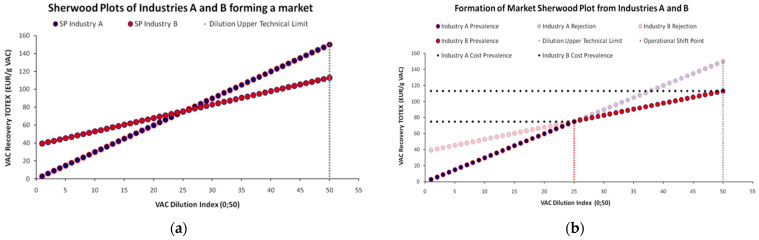
(**a**) Schematic (theoretical) superposition of the SPs of two single industries as a first step of the market formation. *Industry A* from an ontological perspective could be identified as *variable cost intensive* as its constant costs’ coefficient is zero, while *Industry B* could be identified as the *constant cost intensive*, as even for high VAC dilution levels (up to the upper technical limit), its constant costs comprise a large fraction of its total costs; (**b**) The market will seek the minimum cost for the VAC’s recovery at each dilution level; thus it will allocate respectively the VAC’s recovery among industries.

**Figure 11 entropy-25-00004-f011:**
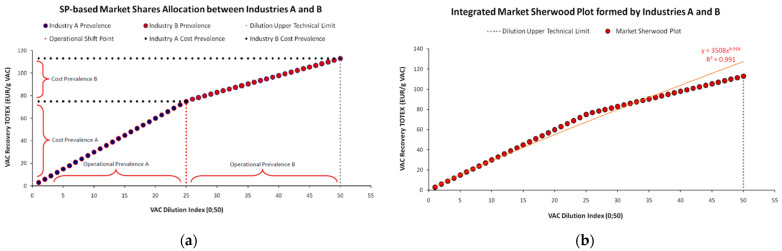
(**a**) Schematic presentation of the *MPI* elements—the *OPI* and the *CPI—*for two (2) hypothetical industries *A*,*B* forming a *VAC recovery market*. It is obvious that although each industry has an *OPI =* 0.5, implying that the market’s control is allocated in equal shares, the *CPI* shows a significantly unequal distribution. For industry *A*, *CPI* = 0.69 (controlling the recovery cost range between 0→78) and for industry *B*, *CP*I = 0.31 (controlling the recovery cost range between 78→113). By the *MPI*, the market relies more on the control of the recovery cost range than of the operational range; (**b**) The integrated *Market Sherwood Plot* accompanied by a non-linear regression. The Market SP shape highly depends on the cost ontologies of the industries forming the market that, in turn, determine the value of the MSDs.

**Table 1 entropy-25-00004-t001:** Classification of main *Cost Categories* (Constant, Variable) and *Classes* (CAPEX, OPEX) with indicative cases and descriptions, depicting the cost structure space of an industry’s *Sherwood Plot* equation.

Cost Category/Class	CAPEX	OPEX
**Constant**	**Infrastructure investment, indirectly related to production volume and recurrent at constant size and periodicity**: Purchase of land, infrastructure (e.g., facilities, machinery) via long-term loans (5–10 years duration) at fixed discount rate or long-term leasing; legal contracts (e.g., company foundation and constitution); product certifications; intellectual property (IP) and patents; equipment insurance.	**Necessary supplies and services for operations, indirectly related to production volume and recurrent at constant size and/or periodicity**: Personnel of strategic hard skills (e.g., R&D, administration) with long-term employment contracts and non-disclosure agreements that is difficult to substitute; resources (e.g., chemicals, fuels, electricity) supplied with forward contracts at fixed prices (e.g., futures); periodical (at fixed dates) equipment maintenance with company personnel or long-term outsourcing contracts.
**Variable**	**Infrastructure investment, directly related to production volume and non-recurrent at variable size and/or periodicity**: Transportation means (e.g., vans, trucks) via variable discount rate loans or cash; marketing and promotion; ad-hoc long-term training programs and certification seminars for strategic personnel.	**Necessary supplies and services for operations, directly related to production volume and non-recurrent at variable size and/or periodicity**: Temporary contractor/outsourced personnel (high or low skilled) of rolling durations that is easy to substitute; ad-hoc personnel short trainings; consumables (e.g., paper, printers); travel costs; ad-hoc equipment maintenance by external personnel at current market prices.

**Table 2 entropy-25-00004-t002:** The *constant* and *variable cost* parameter values as cost categories and their classes (CAPEX, OPEX) forming the TOTEX.

Cost Category/Class	CAPEX	OPEX	TOTEX
**Constant Costs Coefficient *a***	0.0269	0.347	0.347
**Variable Costs Coefficient *b***	12.570	5208.000	5220.570

## Data Availability

The data for extrapolating the case study’s Sherwood Plot are presented aggregated in the categories of *Total Expenditure* (TOTEX), *Capital Expenditure* (CAPEX) and *Operational Expenditure* (OPEX) in Segment 2.4.2 (Figure 5b) and Section 3. For data of higher granularity, special permission from *Greener than Green Technologies S.A.* (https://greenerthangreen.co/) is required. The data on electricity prices for the stochastic models in Section 3 are available for free use and the link to them is provided in [61].

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
