# Peer review of "Resource Recovery and the Sherwood Plot"

_entropy, 2022, doi:10.3390/e25010004_

Round 1
Reviewer 1 Report
This is an interesting work with significant outcomes. I feel you should present the idea better in the introduction- make shorter paragraphs for one so that the reader can identify easier the contribution of your paper. This should also be taken care of in the entire manuscript. Pls make shorter sentecnes, more targeted and clear paragraphs.
Author Response
Response Letter to Reviewer #1
Dear MDPI Entropy journal reviewers, editors and publishing committee,
We deeply thank you for your contribution via your expertise and observations towards the improvement of the scientific accuracy and conveyance of our manuscript’s work significance to the academic community and other audiences who may utilize its value.
The Authors’ Responses (AR) below concern the observations of Reviewer #1 (R#1) as well as the actions adopted for improving our manuscript. Please note that every reference to our manuscript concerns the revised version, submitted along with the current letter.
R#1: This is an interesting work with significant outcomes.
AR: We thank the reviewer for his recognition and credits. To re-introduce the Sherwood Plot to the scientific discussion on Industrial Symbiosis / Circular Economy metrics via accumulating the quite limited related literature and further elaborate on its uncharted potential utilizations at both the physical and economic level was the corner stone of our work.
R#1: I feel you should present the idea better in the introduction- make shorter paragraphs for one so that the reader can identify easier the contribution of your paper. This should also be taken care of in the entire manuscript.
AR: We have implemented significant re-statements in order to incorporate at the maximum degree the reviewer’s suggestions, without sacrificing the highlight of our work’s innovations. We have shortened paragraphs, simplified sentences and have used simpler language across the whole revised manuscript.
R#1: Please make shorter sentences, more targeted and clear paragraphs.
AR: All along the text we have made the necessary changes to simplify as much as possible the wording, shorten the sentences to convey our statements better to the reader and have separated the long paragraphs into shorter segments (2 or 3 in some cases). The above have been applied to all the text of the revised manuscript.
We, the authors, have both agreed on and approved the above responses to Reviewer #1 and have included the respective modifications to our revised manuscript Entropy-1874421 submitted to MDPI Entropy.
We, the authors, are available for any further information and clarification.
Sincerely,
Georgios Karakatsanis and Christos Makropoulos

Reviewer 2 Report
In this manuscript, the authors explained the physical and economic foundations of the Sherwood plot, which is a linear regression of the logarithm of the recovery cost against the logarithm of the dilution factor, before going on to define various generalised Sherwood plots. They then tested these Sherwood plots against real-world data from polyphenol recovery from the waste water generated by fruit juice production, to show that these are valid.
The writing of the paper is clear, but somewhat verbose. More importantly, there are many unnecessary and excessive claims, such as the significance of an increase in the market price of a substance before its recovery is economically feasible from a mixture with high dilution. Such statements are made to sound more profound than they actually are. They should be toned down.
In my opinion, much of Section 1 and Section 2 are not necessary, and can be greatly abbreviated. Section 3, which is where the generalised Sherwood plots are defined, can also be shortened.
For Section 4, I do not understand the use of 'deterministic' versus 'stochastic'. It seems to me that the 'deterministic' data set was purposely prepared to test the Sherwood plots. The procedure for generating this data should be described in detail.
The 'stochastic' data set appears to be electricity prices. These two prices are measured differently, but are highly correlated, so it is not surprising that they fall close to a straight line on a plot of one price against the other. However, can this plot really be called a Sherwood plot? Can one of the prices be interpreted as some kind of dilution factor? It seems to me not.
This is not the only confusing figure. Figure 1(b) shows a scatter plot of different materials in the Sherwood plot. However, why is it that each material appears as a point, at a fixed market price, and a fixed dilution factor? Should not each material be a curve, which is the viable market price for extraction at a given dilution factor?
Section 5 and Section 6 can also greatly shortened. The Conclusions section should also be numbered.
Author Response
Response Letter to Reviewer #2
Dear MDPI Entropy journal reviewers, editors and publishing committee,
We deeply thank you for your contribution via your expertise and observations towards the improvement of the scientific accuracy and conveyance of our manuscript’s work significance to the academic community and other audiences who may utilize its value.
The Authors’ Responses (AR) below concern the observations of Reviewer #2 (R#2) as well as the actions adopted for improving our manuscript. Please note that every reference to our manuscript concerns the revised version, submitted along with the current letter.
R#2: In this manuscript, the authors explained the physical and economic foundations of the Sherwood plot, which is a linear regression of the logarithm of the recovery cost against the logarithm of the dilution factor, before going on to define various generalised Sherwood plots. They then tested these Sherwood plots against real-world data from polyphenol recovery from the waste water generated by fruit juice production, to show that these are valid.
AR: We thank the reviewer for the concise description of our paper and identification of its contribution. To re-introduce the Sherwood Plot to the scientific discussion on Industrial Symbiosis / Circular Economy metrics via accumulating the quite limited related literature and further elaborate on its uncharted potential utilizations at both the physical and economic level was the corner stone of our work. We just point out that in the original version of the Sherwood Plot, the regression concerns the relation between the “dilution” of the target material for recovery and its “required price in the market”. More simply stated, an industry under the original frame of the Sherwood Plot would ask “at what price should the target material be sold in the market for us to consider more economical to process and recover it from a waste matrix instead of mining it or purchase it from a virgin ore?”. This is a part from which our work differentiates as we argue (in segment 2.3.3 of the revised manuscript and in paragraph 2.3 of the old) in favor the adoption of the “dilution to cost of recovery” approach as more fundamental, parsimonious, direct and economically useful. This is an issue that we develop more thoroughly below as the reviewer has more related questions.
R#2: The writing of the paper is clear, but somewhat verbose. More importantly, there are many unnecessary and excessive claims, such as the significance of an increase in the market price of a substance before its recovery is economically feasible from a mixture with high dilution. Such statements are made to sound more profound than they actually are. They should be toned down.
AR: We have implemented significant re-statements in order to incorporate at the maximum degree the reviewer’s suggestions, without sacrificing the highlighting of our work’s contributions. We have shortened paragraphs, simplified sentences and have used less complex language across the whole revised manuscript. From our part is important to denote that as through the process of examining the related literature on the Sherwood Plot and noting down the aspects that we identified to remain unexplored, we set as target for our manuscript to become an original reference for other authors and future works as well. Hence, a degree of exhaustively detailed analysis was considered necessary to enhance our argumentation on something that so far has generally not been developed sufficiently. An indicative example is the connection of the Sherwood Plot to the neoclassical economic theory and the indifference curves (in segment 2.3.1 of the revised manuscript and 2.1 of the old), as a quite original input.
On the example that the reviewer mentions, we agree; to state that if the market price of a highly diluted target material increases it may make economical its recovery, while at the previous lower price its recovery was not, is profound. However, our response to that is that we do not stay only to this conclusion but use it as a bridge to examine a chain of consequences at the micro-economic level, of which the processes we further demonstrate. Besides this obvious a macroscopic effect, it affects the micro-economics of industrial symbiosis clusters; primarily the market structure and allocation (Section 3 of the revised manuscript and Section 5 of the old). Had we stayed only to this statement without further development of its repercussions at the micro-economic level would be indeed redundant and unnecessary.
In conclusion, we agree with the reviewer that there are cases where we have adopted an exhaustively analytical argumentation. In the revised version we limited such cases to where we find an opportunity to verify our argumentation from various angles and not just repeating previous statements.
R#2: In my opinion, much of Section 1 and Section 2 are not necessary, and can be greatly abbreviated.
AR: Sections 1 and 2 of the old manuscript have been incorporated to Section 2 “Materials and Methods” of the revised one. As we aim at substantiating theoretically and empirically the utilization of the Sherwood Plot, we consider necessary to analyze thoroughly both its physical and economic foundations; otherwise the value-added of our work would only be marginal. In any case, according to our general effort to upgrade the style and language of the paper and in the spirit of the conclusion of the reviewer’s previous observation, we have improved and abbreviated the new text.
R#2: Section 3, which is where the generalised Sherwood plots are defined, can also be shortened.
AR: Section 3 in the old manuscript is now incorporated as Paragraph 2.4 in the revised manuscript. As it is already the most mathematically-intensive part with minimum text, it was the most difficult to further abbreviate, as some equations with economic content require further explanations. In any case though, following the reviewer’s suggestions, we have made selected refinements and abbreviations to this part as well.
R#2: For Section 4, I do not understand the use of “deterministic” versus “stochastic”. It seems to me that the “deterministic” data set was purposely prepared to test the Sherwood plots. The procedure for generating this data should be described in detail. The “stochastic” data set appears to be electricity prices. These two prices are measured differently, but are highly correlated, so it is not surprising that they fall close to a straight line on a plot of one price against the other. However, can this plot really be called a Sherwood plot?
AR: Section 4 of the old manuscript has now been incorporated in Section 3 “Results” of the new and specifically the empirical analysis is contained in Paragraphs 3.1 and 3.2. Perhaps the adoption of “fixed data” or “fixed values” instead of “deterministic data” would be more suitable and less confusing. As we describe in the introduction of Section 3 (in the new version), the small pilot unit (Greener than Green Technologies S.A.) provided us data on 5 of its CAPEX and 5 of its OPEX cost elements (10 cost elements in total) for a range of VAC dilutions and with specific calibration of their equipment. Indicative cost elements concerned the cost of personnel, the cost purchasing solvents, the cost of electricity etc.. After these measurements, we were able to extrapolate the unit’s Sherwood Plot for 41 VAC dilution levels from 500mg-100mg/L with a bin (interval) of 10mg/L.
For the “deterministic” data, as the industry provided fixed values on the cost for each of the 10 cost elements -that add up to the total cost for each VAC dilution level- the measurements provided a perfect linear relationship (R2=1) between the VAC’s dilution level and the cost for its recovery; hence, giving a “deterministic” Sherwood Plot, without randomness and deviations around the regression line -as is usually the case of linear regressions and as presented in Fig. 1a of the revised manuscript. As we demonstrate in Fig. 5b for each VAC dilution increase by 10mg/L the total cost for its recovery increases by a factor of 5.220 -depicting more man-hours, more solvents, more electricity consumption etc., as the higher is the VAC’s dilution the higher is the effort to recover it. This is the core assumption of the Sherwood Plot, which was verified.
As the above was useful as an experiment and starting point for the verification of our core assumption on the Sherwood Plot, we needed to expand our measurements into a simulation and add the effect of variability of the 10 cost elements’ purchase prices. Due to the complexity of the task we chose electricity prices as only one of the 10 cost elements to consider that its price is now not fixed but variable. All other 9 cost elements continued to be considered at fixed prices. So, instead of a fixed price of electricity (previously provided by the industry) now we assumed that this is fluctuating around an average value. Hence, for each VAC dilution level the electricity had to be purchased at a different price. Equivalently stated, it is as we assumed that at t1 the industry receives a VAC at dilution level 500mg/L, at t2 it receives the VAC at dilution level 490mg/L etc.. To simulate the variability of the electricity prices we utilized the IPTO daily data between 2019-2020 (365 observations) and estimated its standard deviation (along with other moments as well). Then we incorporated this standard deviation to the industry’s cost of electricity at every VAC dilution level to add randomness to the existing trend. In turn, we examined the effect randomness in electricity as one of the ten cost elements on the industry’s Sherwood Plot. The next step was to iterate the (Monte-Carlo) simulations on the cost of electricity that was found to account for 1,02-1,04% of the total cost. As demonstrated in Fig. 9b the points of the new Sherwood Plot were slightly more “stochastic” than in the previous one with completely fixed data, the total cost factor increase for every increase of dilution by 10mg/L changed to 4.703 and the R2 was slightly lower (=0,99) due to the effect of this small randomness. If we had repeated this rationale for the rest of the 9 remaining cost elements, the Sherwood Plot would be even more stochastic and less “deterministic”.
Finally, as we clearly state in the manuscript, the task of applying the above rationale to all cost elements to incorporate the full variability in the industry’s Sherwood Plot is very demanding and would require a complex econometric analysis, as we would have to examine potential cross-correlations and statistical dependencies between the cost elements (e.g. is the price of solvents statistically depended on the price of electricity? If yes, positively, negatively and at what degree?)- and for various time-lags (t-1, t-2,...,t-n etc.). That of course would require a whole new paper, which is included in our future plans.
R#2: Can one of the prices be interpreted as some kind of dilution factor? It seems to me not.
AR: If we understand correctly the reviewer’s questions, the VAC market prices in the original form of the Sherwood Plot or the cost elements of the industry do not comprise a dilution factor. What we assume in our model is that the industry receives at a specific initial dilution the VAC within the waste matrix and recovers it at a specific total cost -further classified in constant and variable costs. At what total cost the industry is able to recover the VAC depends on how well its production factors collaborate; that part revealed by the values of the Sherwood Plot’s parameters (developed and explained in our mathematical part). In our manuscript we do not claim that either VAC market prices or the costs of production factors affect the dilution of the VAC at the time it is received.
However, only in the context of our discussion with the reviewer we would like to present some more special cases: In the long-term and with a more dynamic view (e.g. with new legislation measures), if the VAC’s market price starts to rise, that could give a signal to the authorities to adopt (technical or other) measures for increasing the concentration of the VAC in the waste matrix during its collection in order to deliver it at lower dilutions to the industries that -in turn- minimize their recovery costs and maximize their profits. Within this dynamic context, in the long-term the market price could also be a factor of the VAC’s dilution. However, this case is completely out of the scope of our current work, concerns a different time-scale and can be the object of a future study.
R#2: This is not the only confusing figure. Figure 1(b) shows a scatter plot of different materials in the Sherwood plot. However, why is it that each material appears as a point, at a fixed market price, and a fixed dilution factor? Should not each material be a curve, which is the viable market price for extraction at a given dilution factor?
AR: As we explain in Paragraph 3 of the “Introduction” in the revised manuscript and Paragraph 2 of the “Introduction” in the old, as well as in the text under the Eq. (2.1) in the revised manuscript and Eq. (1.1) of the old, a fundamental motivation for developing our argumentation on the re-postulation of the Sherwood Plot has to do with what the reviewer accurately points out. The original form of the Sherwood Plot concerns only cross-sectional data (different VACs and different sectors recovering each one of them) that are not sufficiently representative of the market. Fig. 1b depicts the original form of the Sherwood Plot and has been reconstructed with data that we found in the literature (we provide respective references). The main drawback of the original form of the Sherwood Plot is that it is too static. Specifically, each VAC appears as a single point, at a fixed market price and at a fixed dilution level. The existing literature lacks the necessary data from more industries to create a more representative sample, while the reconstructed data concern the reality of 20-30 years ago. We are not even aware if these points were formed by a weighted average or just by the available data that were considered as benchmark at that time. However, it was absolutely necessary to present the original Sherwood Plot as a starting point for our work.
Hence, what we suggest in our work is that ideally, for each VAC in Fig. 1b we should have data from many industries, for more dilution levels and -hence- with more required market prices. Simply stated, for each VAC dilution level we should have a distribution or a range of required market prices, reflecting the different recovery costs of each VAC by a number of industries. That is a more coherent depiction, as each industry is quite different from the other (e.g. in terms of cost efficiency, economies of scale, technology, R&D and innovations) that in turn makes it more or less cost-efficient; thus requiring a different market price to consider the VAC’s recovery economical. With our suggested methodology, we denote the importance of industries keeping track of data at various dilution levels so that we may compare them via statistic/econometric analyses and have a more representative view of the VAC recovery markets.
Finally, in our empirical study we had available data from only one industry for various VAC dilution levels; hence we focused on analyzing this industry’s individual Sherwood Plot. Had we available data from more industries operating in the recovery of the same VAC and for the same dilution levels we would be able to conduct such an empirical study for all industries. Since we didn’t have the available data, we developed the theoretical background of Paragraph 3.3 of the revised manuscript (Section 5 in the old manuscript) on the “Market Structure and the Sherwood Plot”. With this theoretical background formulated we are ready to utilize any future available data for econometric analyses and economic deductions for more than one industry.
R#2: Section 5 and Section 6 can also greatly shortened.
AR: In the revised manuscript, Section 5 (“Market Structure and the Sherwood Plot”) has been incorporated to the Section 3 of the revised manuscript “Results”, while and Section 6 “Research extensions of the Sherwood Plot” has been incorporated to Section 4 “Discussion and Extensions”. Following the reviewer’s recommendations, we have shortened the text and simplified the language. We did this up to a limit where the consistency of our argumentation and highlighting of our contributions to the field would not be sacrificed; taking into consideration that our work aspires to fill a significant gap on the existing literature of the Sherwood Plot’s potential utilizations. Particularly, we reduced significantly -and even removed- parts on “Market Ontologies” in the “Discussion” as this issue concerned a rather philosophical argumentation that was not necessary to develop at such a degree in this work, while we continue to back the main points via related literature references.
R#2: The Conclusions section should also be numbered.
AR: In order to keep the “text” structure of the “Conclusions” section -which we believe is more appropriate to the manuscript’s purposes- but also incorporate the reviewer’s suggestions on making them more clear and straightforward to the reader, we have implemented the following actions: (a) we separated the “Conclusions” section into more paragraphs and (b) have numbered the main points in order to present their logical sequence and highlight the relations between them.
We, the authors, have both agreed on and approved the above responses to Reviewer #2 and have included the respective modifications to our revised manuscript Entropy-1874421 submitted to MDPI Entropy.
We, the authors, are available for any further information and clarification.
Sincerely,
Georgios Karakatsanis and Christos Makropoulos

Round 2
Reviewer 2 Report
Unfortunately, the authors' reply to my comments were just as verbose as the way as they wrote their manuscript. I think that most of my comments are straightforward to address, requiring responses no longer than 3-5 sentences. Instead, for some of these comments the replies are 3-5 paragraphs. More importantly, when I read these long replies, I do not see how they are replies to my comments. I am tempted to recommend another rewrite. However, this does not sound justified, given that I do not understand the replies. Therefore, can I request the authors to reply to my comments again, this time with short and concise sentences?
Author Response
Dear MDPI Entropy journal reviewers, editors and publishing committee,
This is our second version of answers to Reviewer #2. Having read the reviewer’s latest comments, we would like to ensure him/her that our intention was to be analytical in order to leave no unexplained parts. However, as the reviewer was unsatisfied and asked for short and concise answers, we answer again in this style.
R#2: In this manuscript, the authors explained the physical and economic foundations of the Sherwood plot, which is a linear regression of the logarithm of the recovery cost against the logarithm of the dilution factor, before going on to define various generalised Sherwood plots. They then tested these Sherwood plots against real-world data from polyphenol recovery from the waste water generated by fruit juice production, to show that these are valid.
AR: The original Sherwood Plot asks “what is the required market price of a target material for its recovery from a waste matrix to be considered economical?”. Our work restates this issue as “what is the relation between a target material’s dilution in a waste matrix and the cost of its recovery?”. In the manuscript we develop analytically the difference between the two approaches and why our restatement is more straightforward and useful.
R#2: The writing of the paper is clear, but somewhat verbose. More importantly, there are many unnecessary and excessive claims, such as the significance of an increase in the market price of a substance before its recovery is economically feasible from a mixture with high dilution. Such statements are made to sound more profound than they actually are. They should be toned down.
AR: In general, we have shortened paragraphs, simplified sentences and have used less complex language across the whole revised manuscript. On the example that the reviewer mentions (if the market price of a highly diluted target material increases it may make economical its recovery, while at the previous lower price its recovery was not) we use this statement only as a bridge to examine its many dimensions and consequences at the micro-economic level at next sections (Micro-economic foundations, market formation process).
R#2: In my opinion, much of Section 1 and Section 2 are not necessary, and can be greatly abbreviated.
AR: These sections have been incorporated to “Materials and Methods” of the revised manuscript. We simplified the style and language of the paper and have abbreviated the new text without sacrificing the necessary points that require more analysis.
R#2: Section 3, which is where the generalised Sherwood plots are defined, can also be shortened.
AR: Section 3 is the most mathematically-intensive part and it was the most difficult to further abbreviate, as some equations with economic content require further explanations. However, we have made selected refinements and abbreviations to this part as well.
R#2: For Section 4, I do not understand the use of “deterministic” versus “stochastic”. It seems to me that the “deterministic” data set was purposely prepared to test the Sherwood plots. The procedure for generating this data should be described in detail. The “stochastic” data set appears to be electricity prices. These two prices are measured differently, but are highly correlated, so it is not surprising that they fall close to a straight line on a plot of one price against the other. However, can this plot really be called a Sherwood plot?
AR: As described in the manuscript, the small unit provided us data on 5 of its CAPEX and 5 of its OPEX elements. These 10 cost elements add up to the total cost of recovery TOTEX. Initially (deterministic approach), each cost element had a fixed value and was measured for a range of VAC dilutions from 500mg-100mg/L with a bin (interval) of 10mg/L. In total, the unit’s owner measured the cost of VAC recovery for 41 different dilutions. Based on these measurements we extrapolated the unit’s Sherwood Plot, which was a perfect line (R2=1). The basic Sherwood Plot hypothesis was verified (TOTEX increases as dilution increases).
We then tested a slightly more complex model (stochastic approach). In just 1 of the 10 cost elements -electricity- we assumed variable cost. So, after analyzing the statistical properties of the “Clearances Marginal Price” (CMP) time-series in the electricity market of Greece, we incorporated them as part of the TOTEX. The unit’s owner repeated the 41 measurements but this time, at every dilution level the cost of electricity CMP (and only that) was changing. That had a slight effect on the TOTEX making it more random (R2=0,99).
Finally, as we state, the task of applying the above rationale to all 10 cost elements for incorporating randomness is very demanding and would require a complex econometric analysis in a new paper.
R#2: Can one of the prices be interpreted as some kind of dilution factor? It seems to me not.
AR: If we understand correctly the reviewer’s questions, neither the VAC market prices in the original form of the Sherwood Plot nor the cost of the VAC’s recovery in our suggested version of the Sherwood Plot comprise a dilution factor.
R#2: This is not the only confusing figure. Figure 1(b) shows a scatter plot of different materials in the Sherwood plot. However, why is it that each material appears as a point, at a fixed market price, and a fixed dilution factor? Should not each material be a curve, which is the viable market price for extraction at a given dilution factor?
AR: As we explain, Fig. 1b depicts the original form of the Sherwood Plot and has been reconstructed with data that we found in the literature (see reference). This is just the available data we found in the literature on the original form of the Sherwood Plot, at a fixed market price and a fixed dilution factor. We argue that this is a drawback, as ideally, for each VAC we should have data for more dilution levels and -hence- for more market prices. In addition, we should have such samples from by a number of industries to compare their efficiency differences. The Sherwood Plot can be linear or a curve (e.g. exponential or n-root). That depends on the value of the exponents in Eq. (2.13).
R#2: Section 5 and Section 6 can also greatly shortened.
AR: We have shortened the text and simplified the language in these sections. We even removed parts on “Market Ontologies” in the “Discussion” as they concerned philosophical arguments.
R#2: The Conclusions section should also be numbered.
AR: We separated the “Conclusions” section into more paragraphs and have numbered the main points in order to present their sequence and relations between them.
We, the authors, have both agreed on and approved the above responses to Reviewer #2 and have included the respective modifications to our revised manuscript Entropy-1874421 submitted to MDPI Entropy.
We, the authors, are available for any further information and clarification.
Sincerely,
Georgios Karakatsanis and Christos Makropoulos
